# Release of Histone H3K4-reading transcription factors from chromosomes in mitosis is independent of adjacent H3 phosphorylation

Rebecca J. Harris [1], Maninder Heer [1], Mark D. Levasseur[1], Tyrell N. Cartwright [1], Bethany Weston[1], Jennifer L. Mitchell [1], Jonathan M. Coxhead [1], Luke Gaughan[2,3], Lisa Prendergast[1], Daniel Rico[1,3,4,5] ✉ & Jonathan M. G. Higgins [1,3,5] ✉

Histone modifications influence the recruitment of reader proteins to chromosomes to regulate events including transcription and cell division. The idea of a histone code, where combinations of modifications specify unique downstream functions, is widely accepted and can be demonstrated in vitro. For example, on synthetic peptides, phosphorylation of Histone H3 at threonine-3 (H3T3ph) prevents the binding of reader proteins that recognize trimethylation of the adjacent lysine-4 (H3K4me3), including the TAF3 component of TFIID. To study these combinatorial effects in cells, we analyzed the genome-wide distribution of H3T3ph and H3K4me2/3 during mitosis. We find that H3T3ph anti-correlates with adjacent H3K4me2/3 in cells, and that the PHD domain of TAF3 can bind H3K4me2/3 in isolated mitotic chromatin despite the presence of H3T3ph. Unlike in vitro, H3K4 readers are still displaced from chromosomes in mitosis in Haspin-depleted cells lacking H3T3ph. H3T3ph is therefore unlikely to be responsible for transcriptional down-regulation during cell division.

Chromatin undergoes dramatic changes to facilitate cell division. Interphase chromosomes are organized in loops, topologically associating domains (TADs), and compartments that may allow cell-type-specific gene expression programs to be carried out[1]. However, these structures are rapidly lost as cells enter mitosis and the chromosomes condense[2–10]. While many proteins are recruited to chromatin to regulate chromosome condensation, cohesion, and segregation, others are displaced during mitosis. For example, numerous transcription factors lose sequence-specific associations with chromosomes and transcription is minimized, and the recruitment of DNA damage response proteins is curtailed[7–20]. As cells re-enter G1, interphase structures progressively reform and transcription factors must re-

associate to re-initiate gene expression. These findings raise a number of important questions, including how the association of transcription factors with chromatin is regulated during the cell cycle, and how appropriate gene expression patterns are re-established after cell division. Knowledge of how these changes are controlled is vital to understand how accurately chromosome segregation takes place and how interphase functions are properly restored.

In one view, the transmission of transcription factors from mother to daughter cells, even if they are dissociated from chromatin, might be sufficient to transfer memory of the gene transcription program[21]. Nevertheless, it is widely believed that chromatin-based mechanisms provide a means to regulate gene expression during cell division. In

[1]Biosciences Institute, Faculty of Medical Sciences, Newcastle University, Framlington Place, Newcastle Upon Tyne NE2 1HH, UK. [2]Translational and Clinical Research Institute, Faculty of Medical Sciences, Newcastle University, Framlington Place, Newcastle Upon Tyne NE2 1HH, UK. [3]Newcastle University Centre for Cancer, Faculty of Medical Sciences, Framlington Place, Newcastle Upon Tyne NE2 1HH, UK. [4]Centro Andaluz de Biología Molecular y Medicina Regenerativa (CABIMER), CSIC-Universidad Sevilla-Universidad Pablo de Olavide-Junta de Andalucía, 41092 Seville, Spain. [5]These authors jointly supervised this work: Daniel Rico, Jonathan M. G. Higgins. ✉e-mail: daniel.rico@cabimer.es; jonathan.higgins@ncl.ac.uk

this scenario, although many transcription factors may be displaced, so-called bookmarks carried on chromosomes in mitosis contribute to the inheritance of gene expression patterns, or to the speed at which gene transcription is re-initiated. Indeed, the increased DNA accessibility of active promoters is largely maintained in mitosis[6,7,9,10,18,22–25]. Various bookmarks have been proposed, including the retention of transcription factors at a subset of specific binding sites, continued basal levels of transcription, and post-translational modifications of DNA and histones[26,27].

Histone modifications constitute an appealing system to provide a bookmarking mechanism. Many such modifications are implicated in the regulation of gene expression during interphase. While these major regulatory modifications are often recognized and dynamically bound by reader proteins, the modifications themselves are often more stable, with some persisting throughout the cell cycle[6,28–33]. One example is H3K4me3, a mark characteristic of active promoters[10,15,28–31,33,34]. Other histone modifications are relatively dynamic and some, including numerous phosphorylation marks, are strongly upregulated during mitosis. Examples include phosphorylation of H3T3, H3S10, H3S28, and H2AT120[26,35]. One function of these phosphorylation marks is the recruitment of proteins to mitotic chromosomes. For instance, H3T3ph and H2AT120ph bring the Chromosomal Passenger Complex (CPC) to centromeres[36–39].

Histone modifications that are retained from interphase through mitosis could allow local chromatin states to be transmitted through cell division[6,10,15,28–33,40]. Whether reader proteins that recognise these histone marks are also required for bookmarking is less clear. For example, the acetylated-histone reading protein BRD4 is retained on chromatin in mitosis and was originally proposed to be involved in bookmarking, but more recent work suggests that it is the underlying histone marks that are critical[32,41,42]. Consistent with this, numerous proteins that read the histone methylation status at H3K4, H3K9, and H3K27 appear to be substantially displaced from chromosomes in mitosis[19,20,43–46]. Among these proteins is TAF3, a subunit of the general transcription factor complex TFIID, which recognizes H3K4me2/3 (characteristic of active promoters) through its PHD finger[47–50]. TFIID is a component of the RNA polymerase II preinitiation complex and essential for transcription. Because of this, the displacement of TFIID from H3K4me2/3-marked histones during mitosis has been associated with the general downregulation of transcription during cell division[46,49,51].

The striking observation that many major regulatory histone modification sites (e.g. H3K4, H3K9, H3K27, and H2AK119) are adjacent to sites that can be phosphorylated in mitosis led Fischle et al. to propose the existence of methyl-phos switches[52]. In this model, phosphorylation would prevent the proteins that bind to the adjacent methylated lysine from associating with chromatin. Indeed, subsequent work suggested that H3S10 phosphorylation by Aurora B displaces the H3K9me3-binding Heterochromatin Protein 1 (HP1) from chromatin in mitosis[43,44]. It was proposed that methyl-phos switches may be a widespread mechanism to regulate chromatin association of histone reading proteins.

A significant example is H3K4me3 and the adjacent H3T3, a residue that is phosphorylated during mitosis (and meiosis) by the kinase Haspin[35,53]. In the methyl-phos switch model, H3T3 phosphorylation in mitosis would displace reader proteins that bind H3K4me3 at promoters, thus contributing to transcriptional repression, while preserving a memory of gene activation because the H3K4me3 mark persists throughout mitosis. In line with this, in vitro experiments show that more than thirty H3K4 reader proteins, including TAF3[46,49,54,55], can no longer bind to an H3 peptide that also harbors H3T3ph (see Supplementary Table 1). Methyl-phos switching was therefore proposed to be a major mechanism to regulate transcription during cell division and could modulate protein dissociation from chromatin in a locus-specific manner.

This model, however, is based largely on in vitro observations. Notably, methyl-phos switching requires that methylation and phosphorylation co-occur in vivo, but little is known about the precise genome-wide localization of histone phosphorylation in mitosis. Previous studies provided some evidence for H3T3/K4 switching in cells[45,46,49,56–58], but often rely on the inverse correlation of H3T3ph and H3K4 reader protein levels on chromatin in microscopy experiments. Here, we analyze the genome-wide distribution and function of H3T3ph and find that the release of H3K4-reading transcription factors from chromosomes in mitosis is independent of methyl-phos switching.

## Results

### Specificity of antibodies recognizing H3T3ph and H3K4me2/3

We wished to determine the localization of H3T3ph genome-wide in mitosis using chromatin immunoprecipitation and next-generation sequencing (ChIP-seq), and to compare it to H3K4me3. As H3T3 and H3K4 are adjacent residues in Histone H3, it was critical to use antibodies specific for the histone marks in question that are not blocked by each other or by additional modifications in nearby amino acids. Therefore, we characterized the specificity of H3T3ph and H3K4me3 antibodies before performing ChIP-seq experiments.

Previously, we have found that the affinity-purified rabbit polyclonal antibody B8634 selectively binds H3T3ph in vitro and in cells, and that RNAi, inhibition, or CRISPR/Cas9 knockout of the H3T3 kinase Haspin eliminates epitope recognition in cells[35,59,60]. Using ELISAs on chemically synthesized Histone H3 peptides, we found that the B8634 antibody was indeed specific for H3T3ph, and did not recognize H3S10ph, H3T11ph, or H3T22ph (Supplementary Fig. 1A). Importantly, peptides carrying H3K4me1, K4me2, or K4me3 in addition to H3T3ph were recognized similarly to peptides carrying H3T3ph alone, showing that adjacent H3K4 methylation did not block recognition of H3T3ph (Fig. 1A).

Likewise, the rabbit monoclonal H3K4me3 antibody C42D8 did not recognize H3K9, H3K14, H3K23, or H3K27 trimethylation (Supplementary Fig. 1B), but it did bind to both H3K4me2 and H3K4me3 peptides, and recognition of H3K4me2/3 was not significantly altered by adjacent H3T3ph (Fig. 1B) Extensive assessment of C42D8 by others confirms these results[61,62].

Because there is evidence that testing antibodies with histone peptides does not fully recapitulate their recognition properties in ChIP-seq experiments[62], we tested them using recombinant nucleosomes containing H3T3ph, H3K4me3, or both marks together. Again, the antibodies showed specificity for the expected single modifications, and both similarly recognized the H3T3phK4me3 dually-modified nucleosomes (Fig. 1C, D). We also confirmed that the H3T3ph antibody recognized H3T3ph and H3T3phK4me3-containing nucleosomes equally in the presence of sheared mitotic chromatin, conditions closely approximating those of a ChIP-seq experiment (Fig. 1E).

Finally, we used immunoblotting to confirm that the B8634 H3T3ph and C42D8 H3K4me2/3 antibodies recognized predominantly an H3-sized protein of approximately 17 kDa in HeLa cell lysates, and that H3T3ph was detected in mitotic but not asynchronously growing cells, whereas H3K4me2/3 was found at similar levels in both cases (Supplementary Fig. 1C). From these results we conclude that the B8634 H3T3ph antibody recognizes H3T3ph specifically whether or not H3K4 is methylated, and that the C42D8 antibody is suitable to monitor H3K4me2/3 regardless of the presence of H3T3ph.

### Genome distribution of H3T3ph and H3K4me2/3 in interphase and mitosis

To determine the distribution of H3T3ph and H3K4me2/3 in mitosis using ChIP-seq, we required highly enriched populations of mitotic cells. We used HeLa cells because this is a well-characterized cell line

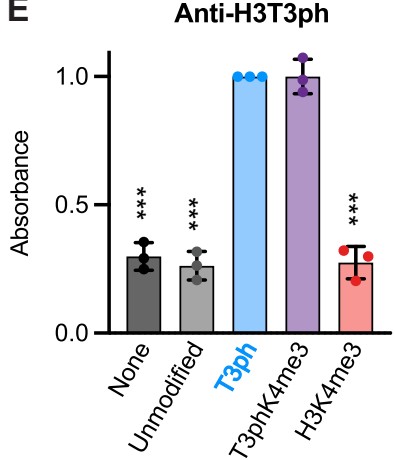

**Fig. 1 | Antibody characterization by peptide and nucleosome ELISA. A** H3T3ph antibody B8634 binding to various H3 peptides (for no peptide *n* = 5; for unmodified *n* = 7; for T3ph and T3phK4me3 *n* = 8; for T3phK4me1 and T3phK4me2 *n* = 3; for K4me3 *n* = 4). **B** H3K4me3 antibody C42D8 binding to various H3 peptides (for no peptide, unmodified and T3phK4me2 *n* = 4; for K4me1 *n* = 5; for K4me2, K4me3, and T3phK4me3 *n* = 6; for T3phK4me2 *n* = 3). **C** H3T3ph antibody B8634 binding to peptide and nucleosomal substrates (*n* = 3). **D**. H3K4me3 antibody C42D8 binding to peptide and nucleosomal substrates (*n* = 3). **E** H3T3ph antibody B8634 binding to nucleosomes in the presence of sheared chromatin (*n* = 3). Data were normalized to the mean signal of the antibodies binding the expected target peptide or nucleosome (i.e. H3T3ph or H3K4me3). Bars represent mean ± SD; *n* refers to the number of independent ELISA experiments. Statistical analysis was carried out with non-normalized data using a mixed effects model with Dunnett's adjustment for multiple comparisons (**A**, **B**), or by a repeated measures one-way ANOVA with Šídák's adjustment for multiple comparisons (**C**–**E**), ***$p < 0.0001$, **$p < 0.001$, *$p < 0.01$ when compared to binding to the expected modification (H3T3ph or H3K4me3). Source data including exact p values are provided as a Source Data file.

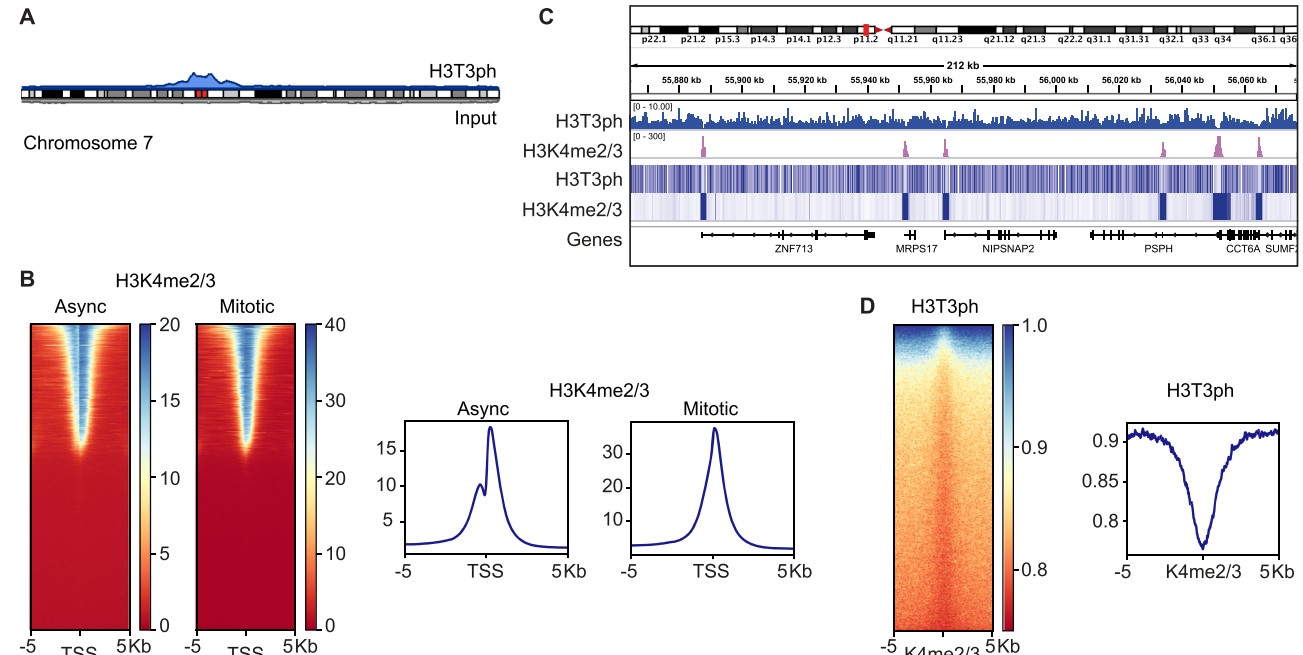

**Fig. 2 | ChIP-seq for H3T3ph and H3K4me2/3 in HeLa cells.** Asynchronous and mitotic HeLa cells were prepared as described in Supplementary Fig. 2. **A** Ideogram showing enrichment of H3T3ph in mitosis at the centromere of human chromosome 7. H3T3ph (blue) and input (gray) read data from a single replicate were aligned to GRCh38.p12. Ideograms for all chromosomes are shown in Supplementary Fig. 3A. **B** Genome-wide analysis of H3K4me2/3 enrichment at transcription start sites (TSS). Heatmaps (left) and metagene plots (right) showing H3K4me2/3 ChIP-seq of asynchronous (Async) and mitotic-enriched cells across 10 kb regions centered at TSSs. **C** Integrative Genomics Viewer (IGV) tracks of mitotic H3T3ph and H3K4me2/3 ChIP-seq showing alignment on a 212 kb region of chromosome 7, represented by read coverage (input normalized) (middle panel) or heatmaps (bottom panel). **D** Heatmap (left) and metagene plot (right) of H3T3ph ChIP-seq at 10 kb regions centered on mitotic H3K4me2/3 ChIP-seq peaks. Vertical scales show enrichment scores. Representative replicate data are shown in Supplementary Fig. 3B, C, and H3T3ph results split into centromeric and non-centromeric regions are shown in Supplementary Fig. 4.

that has been used for a number of comparable studies. We found that a double thymidine block to synchronize cells at the G1/S boundary, followed by incubation with the microtubule poison nocodazole from 8 to 13 h post-release and a mitotic shake-off step routinely enriched mitotic cells to over 90% (as determined by flow cytometry with DNA and MPM-2 staining; Figure Supplementary Fig. 2 A, B). We also confirmed that the level of H3T3ph was largely preserved in preparations of sheared chromatin from mitotic cells compared to whole HeLa cell lysates (Supplementary Fig. 2C). We then carried out ChIP-seq for H3T3ph and H3K4me2/3 on asynchronously growing and mitotically-enriched cell populations. As expected, we recovered DNA from H3K4me2/3 ChIPs from both asynchronous and mitotic cells and from H3T3ph ChIPs from mitotic cells, but little DNA was recovered for H3T3ph from asynchronous cells and these samples were not sequenced.

In previous immunofluorescence studies, H3T3ph emerges in early mitosis on chromosome arms, and becomes enriched around centromeres as prometaphase progresses[35,63]. Consistent with this, alignment of H3T3ph ChIP-seq data from nocodazole-arrested mitotic cells with the human genome revealed strong enrichment in the centromeric regions of all chromosomes (Fig. 2A and Supplementary Fig. 3A, B). As previously reported, H3K4me2/3 was found predominantly at promoters in both asynchronous and mitotic cells (Supplementary Fig. 3C). In asynchronous cells, H3K4me2/3 flanked the nucleosome-depleted regions (NDRs) at promoters of active genes but, in mitosis, we observed a clear loss of NDR signals (Fig. 2B), presumably due to nucleosomes spreading into the NDRs following dissociation of transcription factors as previously described[10,15,25,30,34].

H3T3ph was enriched around centromeres, but the regions containing high levels of H3T3ph extended into euchromatic regions containing genes with H3K4me2/3 at their promoters (Fig. 2A, C). Strikingly, H3T3ph appeared to be selectively depleted at such centromere-proximal promoters (Fig. 2C). Indeed, when we examined the enrichment of H3T3ph around H3K4me2/3 peaks in mitosis, either genome-wide or only in centromere-proximal regions, the phosphorylation mark clearly declined in these regions (Fig. 2D; Supplementary Fig. 4). Only 0.3% of annotated transcription start sites with H3K4me2/3 peaks had detectable H3T3ph and, although the corresponding genes were enriched around centromeres as expected, they did not fall into distinctive functional categories, nor were they enriched among the genes that change expression most clearly between interphase and mitosis in HeLa cells as determined by nascent RNA-seq[15]. These results suggest that the presence of H3K4me2/3 prevents the deposition of H3T3ph on adjacent sites in cells in mitosis.

## Phosphorylation of H3T3 is inhibited on nucleosomes carrying H3K4me3

A simple explanation for the absence of H3T3ph on mitotic chromatin in the vicinity of H3K4me2/3 would be if these methylation marks hinder the activity of H3T3 kinases. Indeed, previous work from our laboratory, and others, has shown that the activity of the mitotic H3T3 kinase Haspin on H3 peptides is progressively inhibited by mono, di, and trimethylation of H3K4[64–66]. Phosphorylation of H3K4me3-containing peptides was severely curtailed, while phosphorylation of H3K4me2 peptides was diminished 2 to 3-fold. We confirmed that this was also true for H3 in nucleosomes, where H3K4me3 caused a clear reduction in the ability of Haspin to phosphorylate H3T3ph (Fig. 3A). Although there is strong evidence that Haspin is the major H3T3 kinase in mitotic cells[35,59,60,63,67,68], other potential kinases have been proposed[69,70]. However, we found that phosphorylation of H3T3 driven by all endogenous kinases active in lysates of mitotic HeLa cells was severely curtailed by H3K4me3 (Fig. 3B). Furthermore, the detected H3T3 kinase activity was strongly inhibited by the Haspin inhibitor 5-iodotubercidin[59,67], and so was ascribable to Haspin (Fig. 3C).

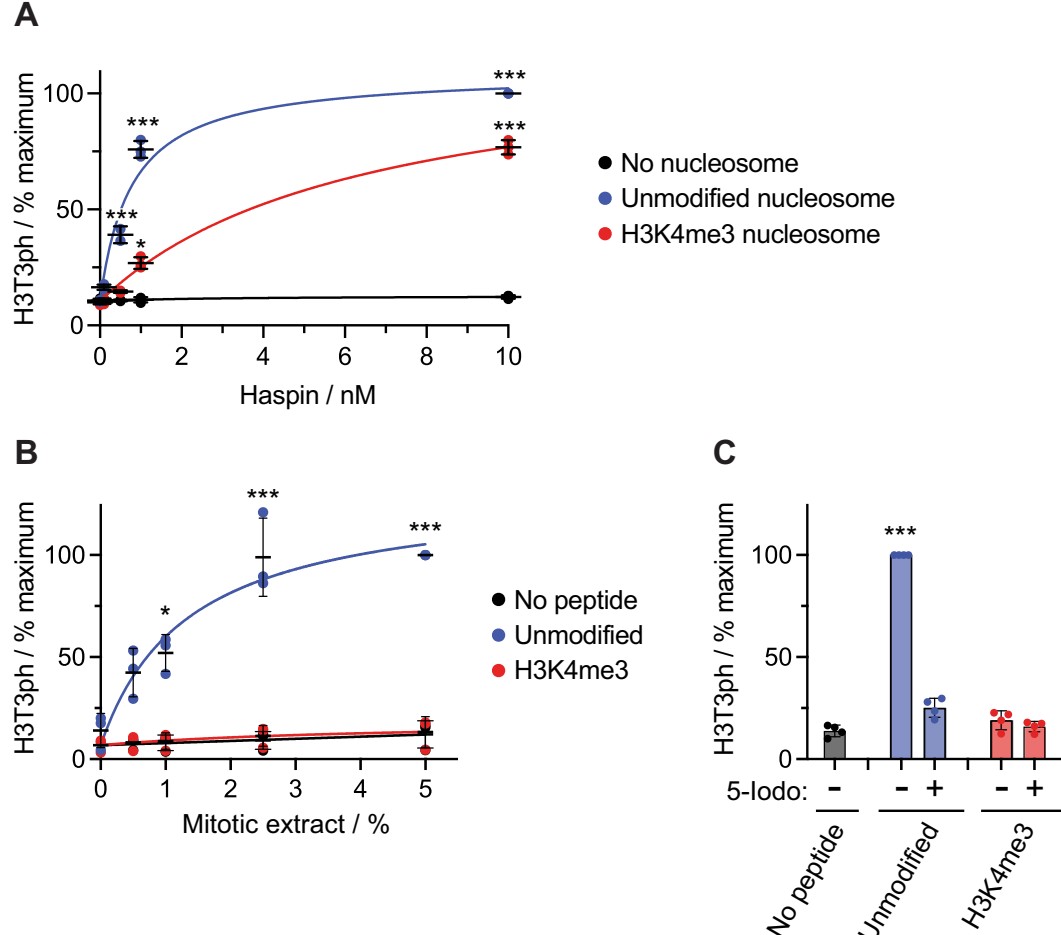

**Fig. 3 | Haspin kinase activity towards H3T3 is inhibited by H3K4me3. A** In vitro kinase assay of recombinant human Haspin using recombinant nucleosomes with the indicated modifications as substrates ($n = 3$). ***$p < 0.0001$, *$p = 0.0035$, using a mixed effects model with Dunnett's adjustments for multiple comparisons, when compared to phosphorylation in the absence of peptide. **B** Phosphorylation of modified H3 peptides driven by mitotic HeLa cell extracts ($n = 3$). ***$p < 0.0001$, *$p = 0.0058$, using a repeated measures two-way ANOVA with Dunnett's adjustment for multiple comparisons, when compared to phosphorylation in the absence of peptide. **C** Phosphorylation of modified H3 peptides driven by mitotic HeLa cell extracts in the presence and absence of 10 μM Haspin inhibitor 5-iodotubercidin (5-Iodo) ($n = 4$). ***$p < 0.0001$ using a repeated measures one-way ANOVA with Dunnett's adjustment for multiple comparisons, when compared to phosphorylation in the absence of peptide. Bars represent mean ± SD; $n$ refers to the number of independent kinase assays. All statistical analyses were carried out using non-normalized data. Source data including exact $p$ values are provided as a Source Data file.

We conclude, therefore, that mitotic kinases are unable to efficiently phosphorylate H3T3 when the adjacent H3K4 is trimethylated.

**Haspin is not required to displace TFIID from mitotic chromatin**
Previous work has shown that the PHD finger of TAF3 binds to H3K4me3, and to a lesser extent to H3K4me2[47,48,54,55]. The binding sites of TAF3, and the TAF3 PHD finger alone, correlate well with the positions of H3K4me3 across the genome in asynchronous cells, consistent with the idea that H3K4 methylation plays an important role in bringing TAF3 to chromosomes[48,50,71]. In experiments using synthetic H3 peptides in vitro, detectable binding of the TAF3 PHD, and of the TFIID complex containing TAF3, to H3K4me3 is eliminated by adjacent H3T3ph (see below)[46,49,54,55]. However, if deposition of H3T3ph by Haspin in cells is impeded by adjacent H3K4me2/3, then methyl-phos switching is unlikely and displacement of TFIID from chromosomes in mitosis should not be influenced by loss of H3T3ph. To test this, we examined the localization of TFIID components in cells lacking Haspin activity. All major TFIID components co-purify with GFP-TAF5[47]; multiple TAF proteins, including TAF5, are pulled down from cell lysates by H3K4me3-containing peptides[48]; and TAF5 has previously been used to monitor TFIID binding to H3K4me3[49], making it a suitable choice for these experiments.

In immunofluorescence experiments using wild-type HeLa cells, GFP-TAF5 was nuclear in interphase, and rapidly displaced from chromosomes in prophase, as previously reported[49] (Fig. 4A). Indistinguishable results were obtained in HeLa cells that lacked detectable H3T3ph due to CRISPR-Cas9-mediated disruption of the Haspin gene (Fig. 4A). To quantify these results, and to ensure they were not influenced by formaldehyde fixation artefacts[7], we analyzed the location of GFP-TAF5 in live cells throughout mitosis. Removal of GFP-TAF5 from chromosomes began within 5 min of nuclear envelope breakdown, and recovery started approximately 5 min after anaphase onset (Fig. 4B, C; Supplementary Fig. 5A; and Supplementary Movies 1 and 2). This finding was consistent with multiple reports of TAF protein dissociation from mitotic chromosomes using a variety of methods, both with and without the use of formaldehyde[9,20,49,51,72]. Importantly, there was no significant difference in the results in wild type or Haspin knockout cells (two-way mixed effects model ANOVA, $p = 0.30$). Haspin depletion by RNAi also did not detectably influence GFP-TAF5 localization in living or fixed U2OS cells (Supplementary Fig. 5B, C). Upon inhibition of Haspin kinase activity with 5-iodotubercidin in U2OS cells[59,67], we again failed to see any change in GFP-TAF5 displacement by immunofluorescence microscopy (Supplementary Fig. 6A), and similar results were obtained for GFP-TAF3 in HeLa cells

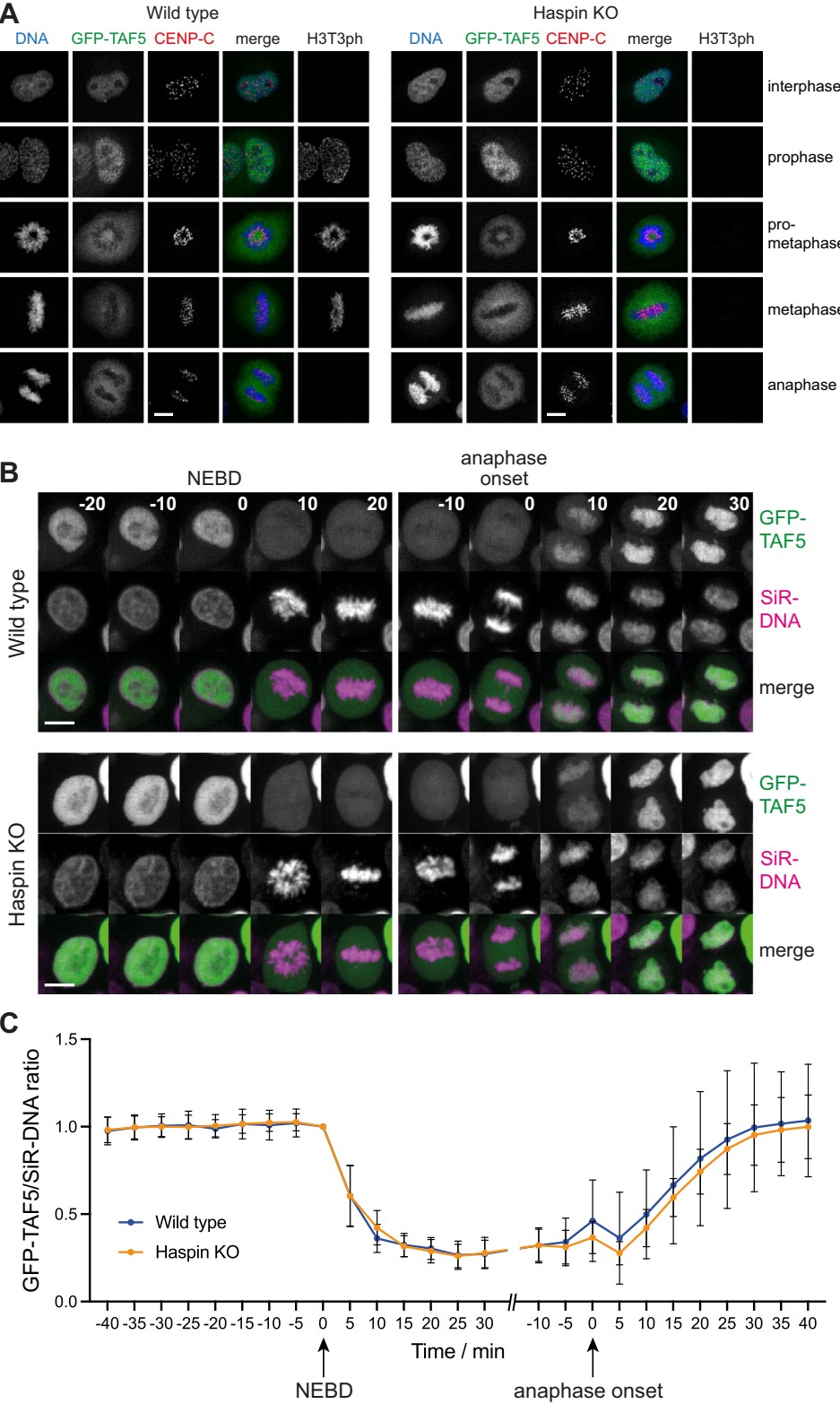

**Fig. 4 | Haspin knock out does not influence the displacement of GFP-TAF5 from chromosomes in mitosis. A** Immunofluorescence microscopy (with formaldehyde fixation) for DNA (blue), GFP-TAF5 (green), CENP-C (centromeres, red), and H3T3ph (gray) in wild type and Haspin knockout (KO) HeLa cells. The experiment was performed once, but the result was confirmed using live imaging (below) and in U2OS cells following Haspin RNAi or Haspin inhibitor treatment (see Supplementary Figs. 5B, C and 6A). **B** Live imaging of GFP-TAF5-expressing wild type and Haspin knockout HeLa cells. DNA was stained with SiR-DNA, images were taken every 5 min, and times are stated in minutes before and after nuclear envelope breakdown (NEBD) and anaphase onset as appropriate. For display purposes only, the intensities of GFP and SiR-DNA images were adjusted separately for each cell to allow visual comparison. Scale bars = 10 μm. **C** Quantification of GFP-TAF5/SiR-DNA ratio during mitosis in live wild type ($n = 9$ cells) and Haspin knockout ($n = 10$ cells) HeLa cells imaged as in **B**, from 3 independent experiments. Bars represent mean ± SD. Individual cell traces are shown in Supplementary Fig. 5A. Source data including exact p values are provided as a Source Data file. See also Supplementary Movies 1 and 2.

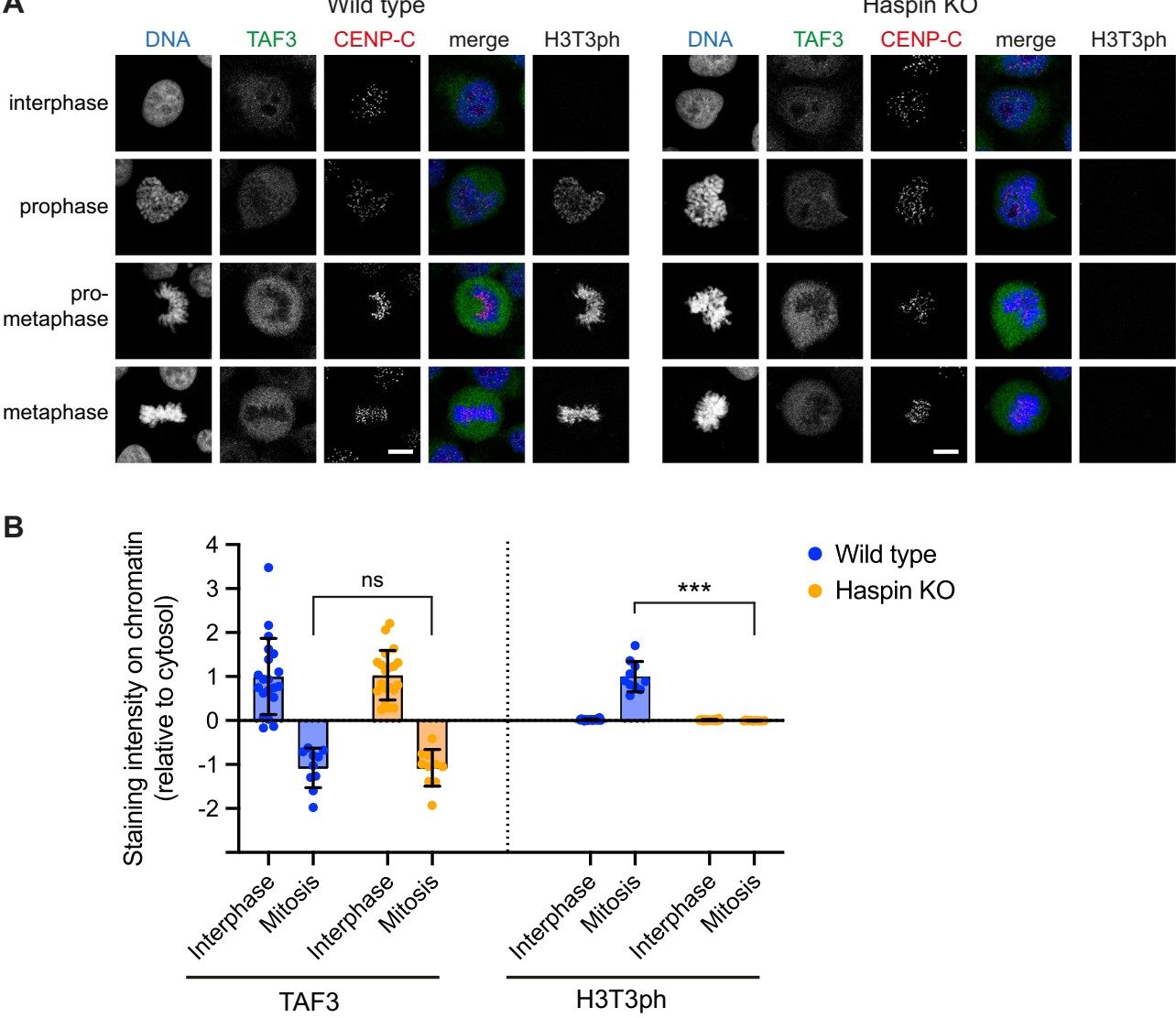

**Fig. 5 | Haspin knockout does not influence the displacement of endogenous TAF3 from chromosomes in mitosis. A** Immunofluorescence microscopy (with formaldehyde fixation) for DNA (blue), TAF3 (green), CENP-C (centromeres, red), and H3T3ph (gray) in wild type and Haspin knockout HeLa cells. The experiment was performed once, but the result was confirmed using GFP-TAF3 and Haspin inhibitor treatment (see Supplementary Fig. 6B). Scale bars = 10 μm. **B** Quantification of results in A. The ratio of TAF3 and H3T3ph staining intensity to DNA staining intensity in chromatin-containing regions, relative to that in the cytosol, was determined. TAF3 intensity was normalized to that in wild type interphase nuclei; and H3T3ph intensity was normalized to that on wild type mitotic chromatin. Bars represent mean ± SD. Statistical analysis was carried out using non-normalized data by unpaired two-tailed Student $t$ tests. ***$p = 3.2 \times 10^{-8}$; ns not significant ($p = 0.99$); $n = 20$ interphase cells and 10 mitotic cells. Source data including exact $p$ values are provided as a Source Data file.

(Supplementary Fig. 6B). Finally, immunofluorescence for endogenous TAF3 gave indistinguishable results in wild type and Haspin knockout HeLa cells (Fig. 5). Therefore, we could not reveal any effect of Haspin activity or H3T3ph on displacement or re-association of TFIID with chromosomes in mitosis.

## The TAF3 PHD finger binds similarly to interphase and mitotic chromatin

The results described above reveal the existence of an H3T3ph-independent mechanism to release TFIID from chromosomes in mitosis, which is likely to involve the phosphorylation of TFIID components by mitotic kinases[51]. However, protein complexes such as TFIID are likely to make multiple contacts with chromatin, and our results so far do not rule out that a methyl-phos switch involving H3T3ph acts redundantly to modulate TAF3 binding to H3K4me2/3. To address this question, we adopted an in vitro approach in which we could determine the chromosome binding sites of the TAF3 PHD finger in the absence of

other TFIID components and their post-translational modifications: chromatin interacting domain precipitation and sequencing (CIDOP-seq)[54]. In this variation of the ChIP-seq method, rather than using an antibody as the probe, we determined the genome-wide binding sites of a recombinant GST-TAF3 PHD finger fusion protein and, as a control, a M880A mutant that is defective in H3K4me2/3 binding[48,54]. Notably, this approach also tests the influence of H3T3ph on TAF3 function in a way that does not rely on knowledge of histone antibody specificity.

We first confirmed that the GST-TAF3 PHD, but not the M880A mutant, was able to bind to H3K4me3-containing peptides in vitro, and that this binding was strongly blocked by adjacent H3T3ph (Supplementary Fig. 7A). As expected, GST-TAF3 PHD (but not the M880A mutant) pulled down H3K4me3-containing histone H3 from sheared chromatin of asynchronous HeLa cells as determined by immunoblotting (Supplementary Fig. 7B). When using chromatin from mitotic cells, GST-TAF3 PHD (but not the M880A mutant) pulled down H3K4me3-containing H3 in similar amounts, and this did not contain

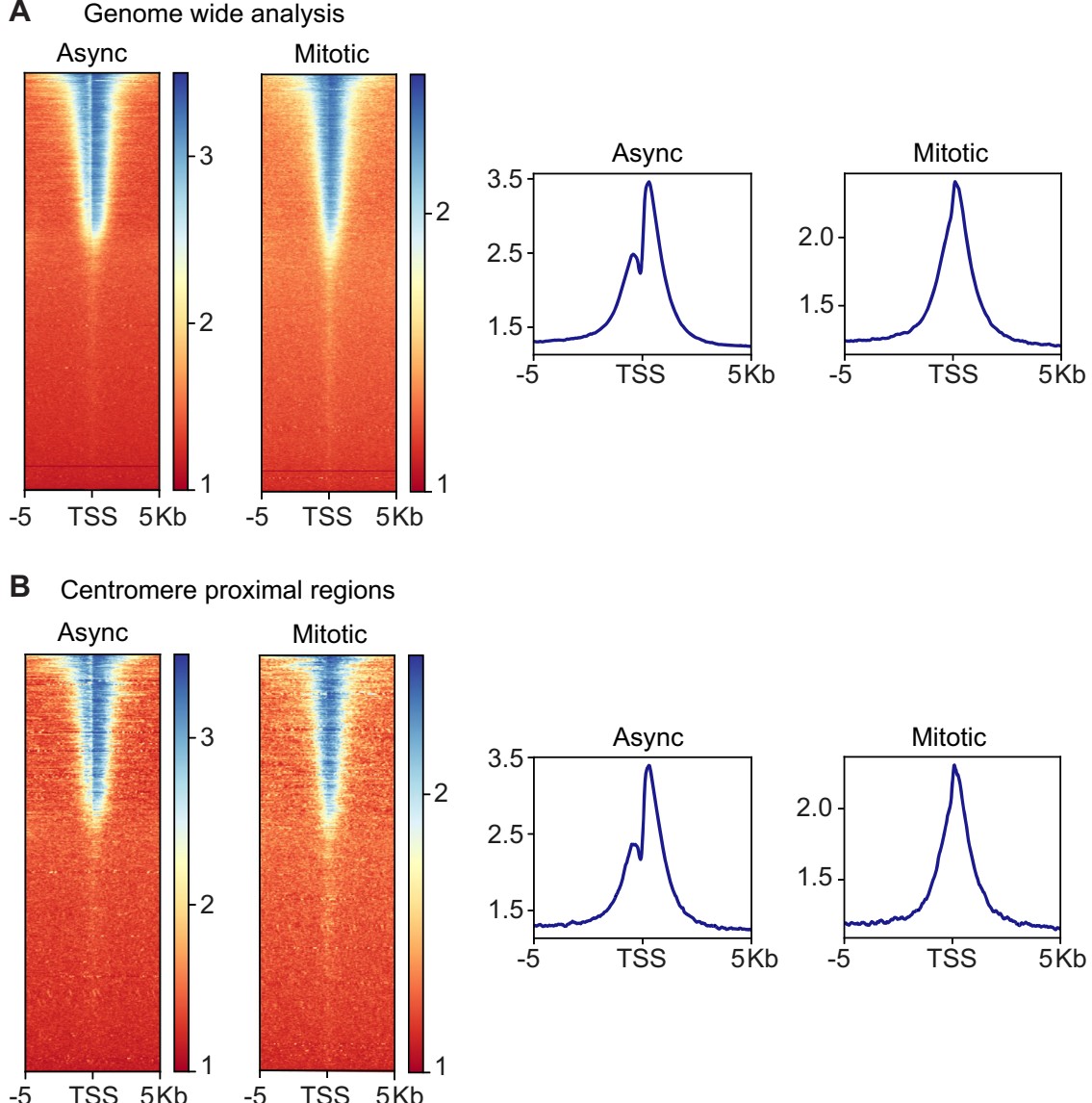

**Fig. 6 | CIDOP-seq shows TAF3 PHD binding to interphase and mitotic chromatin. A** Genome wide analysis of TAF3 PHD enrichment at TSSs. **B** TAF3 PHD enrichment at centromere proximal TSSs. Heatmaps (left) and metagene plots (right) show TAF3-PHD binding across 10 kb regions centered at TSSs for both asynchronous and mitotic-enriched HeLa cell chromatin. Vertical scales show enrichment scores.

H3T3ph (Supplementary Fig. 7B). Therefore, GST-TAF3 PHD can bind to mitotic chromatin, presumably at H3K4me3 sites. The presence of H3T3ph on mitotic chromatin did not detectably reduce GST-TAF3 PHD binding, likely because H3T3ph is not found adjacent to H3K4me2/3 in cells. However, from these results, we could not exclude that H3T3ph was able to prevent GST-TAF3 PHD binding to a subset of H3K4me3-containing regions, such as those surrounding centromeres.

To analyze this in further detail, we carried out CIDOP-seq to identify the genome-wide binding sites of the TAF3 PHD finger in vitro. As previously reported, TAF3 PHD bound to chromatin isolated from asynchronous cells in a pattern similar to that of H3K4me2/3, flanking the NDRs at promoters of active genes[54] (Fig. 6A, async; Supplementary Fig. 8A, B). As expected, M880A mutant TAF3 PHD showed only weak enrichment at H3K4me2/3 sites (Supplementary Fig. 8C). When using mitotic chromatin, GST-TAF3 PHD retained the ability to bind promoters and spread into NDRs, consistent with the incursion of nucleosomes (Fig. 6A, mitotic), as seen for H3K4me2/3 ChIP-seq (see Fig. 3). Importantly, TAF3 PHD was equally enriched at transcription start sites of genes in centromere proximal regions with high overall

levels of H3T3ph (Fig. 6B) as it was genome wide (Fig. 6A), or along chromosome arms where H3T3ph was low (Supplementary Fig. 8D). These results confirmed that H3T3ph does not prevent the interaction of the TAF3 PHD with H3K4me3 on mitotic chromosomes. Therefore, we conclude that, although H3T3ph does displace TAF3 from H3K4me3 peptides in vitro, the absence of H3T3ph immediately adjacent to H3K4me2/3 in cells essentially precludes the operation of a methyl-phos TAF3 switch in vivo.

## Absence of a methyl-phos switch for additional H3K4 reader proteins

Finally, we wanted to test if other reported H3K4 reading proteins are influenced by methyl-phos switching with H3T3ph. ING2 and Dido3 are H3K4me3-reading proteins that no longer bind H3K4me3 in vitro if H3T3ph is adjacent[45,49,73–75]. As previously reported[45,46,76], ING proteins and Dido3 were displaced from chromosomes in mitosis and we found that this was indistinguishable in cells lacking Haspin and H3T3ph, independent of the fixation conditions (Fig. 7A and Supplementary Fig. 9A). Interestingly, the displacement of the H3K4me0-reading LSD1

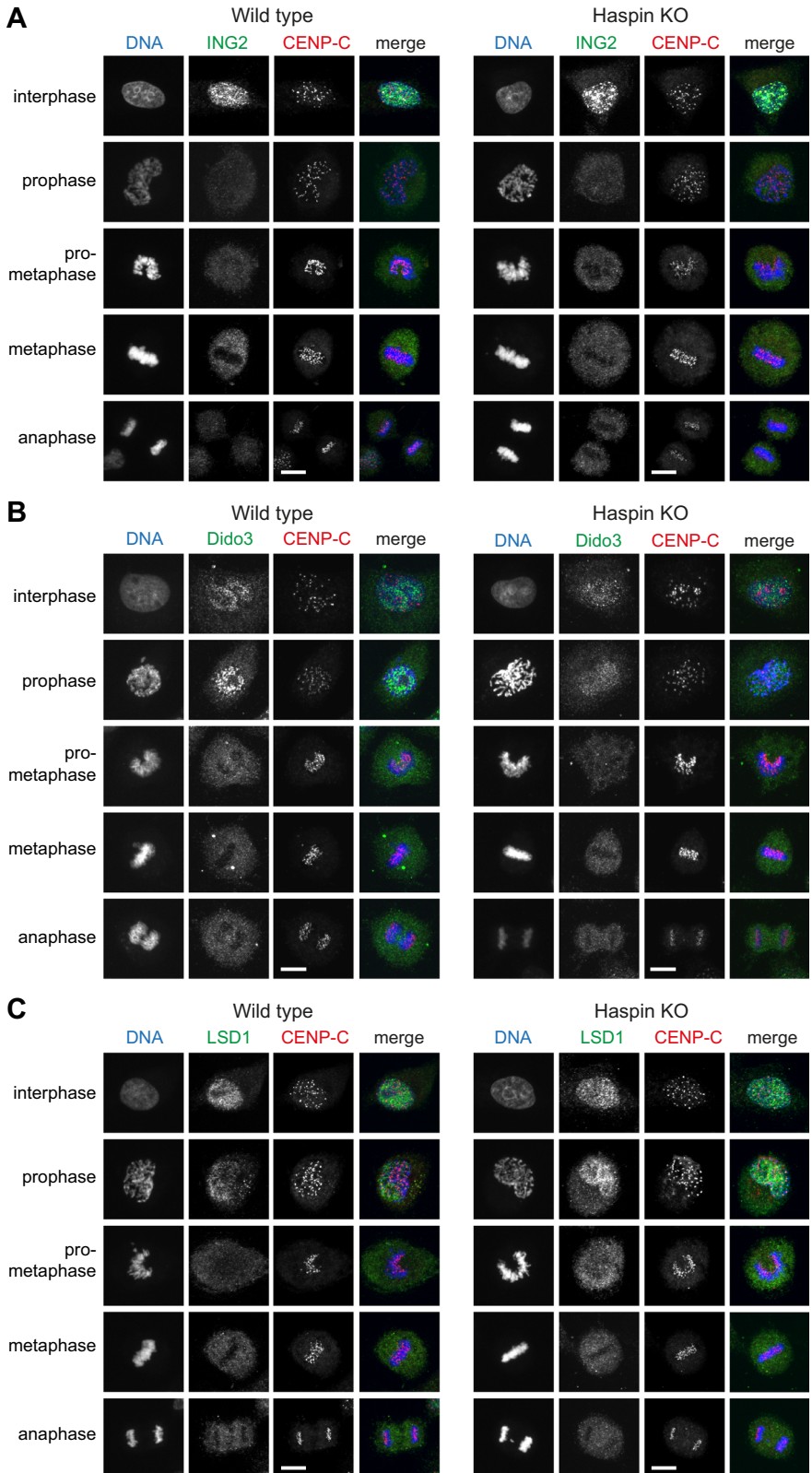

**Fig. 7 | Haspin knockout does not influence the displacement of endogenous ING2, Dido3, or LSD1 from chromosomes in mitosis. A** Immunofluorescence microscopy (with methanol fixation) for DNA (blue), ING2 (green), CENP-C (centromeres, red), and H3T3ph (gray) in wild type and Haspin knockout HeLa cells. Immunofluorescence staining was carried out independently twice with methanol fixation, and 3 times with formaldehyde fixation (see Supplementary Fig. 9A). **B** As for A, but staining for Dido3 (green). Immunofluorescence staining was carried out once. **C** As for A, but staining for LSD1 (green). Immunofluorescence staining was carried out once with methanol fixation, and three times independently with formaldehyde fixation (see Supplementary Fig. 9B). Scale bars = 10 μm.

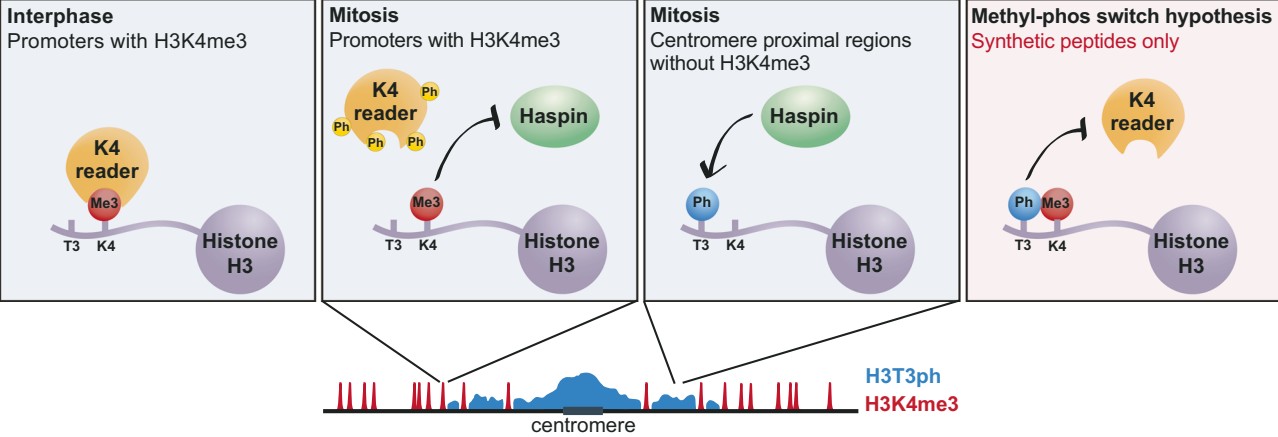

**Fig. 8 | Schematic showing the effects of H3T3ph on H3K4me3-reader binding.** During interphase, H3K4 readers can bind to H3K4me3-methylated histones which are enriched at gene promoters. In mitosis, many H3K4 readers are displaced from promoters, perhaps due to direct phosphorylation of the reader proteins themselves. This release is independent of H3T3ph. Indeed, deposition of H3T3ph by Haspin is inhibited by adjacent H3K4me3. Instead, during mitosis, H3T3ph is deposited in centromere-proximal regions where H3K4me3 is low. On synthetic peptides (red box), H3T3ph does prevent binding of many H3K4 reader proteins, but this rarely happens in vivo.

complex from mitotic chromosomes was also not influenced by loss of H3T3ph (Fig. 7B and Supplementary Fig. 9B), even though detectable binding of the LSD1 complex component BHC80 to H3 peptides is eliminated by H3T3ph in vitro[49,73]. We were, therefore, unable to observe methyl-phos switching at H3T3-H3K4 in cells.

## Discussion

Methyl-phos switching is an appealing concept that has been invoked to explain the dissociation of numerous proteins from chromosomes during cell division. In the case of factors that read the modification state of H3K4, more than thirty reader domains have been found to be displaced by H3T3ph in vitro, and this has been proposed to be important in vivo (see Introduction). Our results, however, suggest that H3T3ph is rarely deposited next to H3K4me2/3 in cells. Therefore, although we cannot rule out that reader binding to a minority of H3K4me2/3 sites is regulated by H3T3ph, global displacement of H3K4me2/3 reader proteins from mitotic chromatin is unlikely to be due to methyl-phos switching (Fig. 8). The chromosomal localization of some multi-domain proteins or protein complexes may not be determined only by H3K4-binding activity, even if they contain H3K4-reading domains[50,77]. H3K4-reading activity probably serves to modulate the conformation and/or function of such proteins in situ. Our results show that this type of H3K4me2/3-reader activity is also unlikely to be regulated by H3T3 phosphorylation.

Previous studies from our group and others have shown that the kinase activity of human and *Arabidopsis* Haspin towards H3T3 within histone peptides is progressively inhibited by mono-, di- and tri-methylation of H3K4[64–66]. Indeed, the crystal structure of Haspin in complex with an H3 substrate peptide suggests that methylation of H3K4 would interfere with phosphorylation of H3T3[78]. We extend this here, showing that H3K4me3 substantially blunts the ability of human Haspin (and, indeed, any kinase active in mitotic cell lysates) to deposit H3T3ph on nucleosomes in vitro. Therefore, we conclude that the diminished H3T3ph in chromatin regions containing H3K4me2/3 is due to blocking of Haspin activity at these sites (Fig. 8). Interestingly, a dip in H3T3ph at active promoters containing H3K4me3 has also been observed in *Arabidopsis* and possibly in *Chlamydomonas*, and H3T3ph is reduced in promoter regions containing active RNA polymerase II (Pol II S5ph) in *Drosophila* S2 cells[79–81]. Although the kinase depositing H3T3ph and/or the cell cycle stage involved was less clear in these studies, the antagonistic effect of H3K4me2/3 on H3T3ph appears to be conserved.

A previous study suggested the existence of a combinatorial mark on histone H3 in mitosis, comprising H3T3ph, H3K4me3, and H3R8me2[63]. However, it is difficult to obtain definitive evidence for a particular combination of marks using polyclonal mixtures of antibodies or MALDI-TOF alone[82,83]. Of course, we cannot rule out that adjacent H3T3ph and H3K4me2/3 marks are present at a low level, but tandem mass spectrometry studies of histone modifications to date have found H3T3ph alone or in combination with H3K4me1, but not with H3K4me2/3[84–87], consistent with our ChIP-seq data.

The results of prior work appeared to support the idea that Haspin and H3T3ph regulate the association of TAF3 with H3K4me3 in cells[49]. A significant finding in this previous study was that overexpression of Haspin could repress TAF3-mediated gene activation. However, Haspin is many-fold overexpressed in these experiments, leading to increased H3T3 phosphorylation in both interphase and mitosis[35,49]. Because high concentrations of Haspin can overcome the suppressive effect of H3K4me3 in vitro (Fig. 3A), and we know that H3T3ph does displace TAF3 from H3K4me3 peptides in vitro[46,49,54,55], it seems likely that elevating Haspin activity was sufficient to artificially flip the methyl-phos switch in this previous work. It is less clear why Haspin RNAi led to GFP-TAF5 retention on mitotic chromosomes in the previous study, in contrast to our current results.

Some previous cell-based studies provided only indirect evidence for H3T3ph/H3K4me3 switching because they relied on the inverse correlation of H3T3ph and H3K4me3 reader protein levels on chromosomes[45,56]. Our results are not at odds with the idea that a number of H3K4me-reading proteins are lost from chromosomes in mitosis, but they argue that H3T3ph is not responsible. Treatment with the Haspin inhibitor CHR-6494 decreased H3T3ph in another report, and this was suggested to cause retention of the H3K4me3-reading proteins Dido3 and ING1 on chromatin during mitosis[46]. However, there was little detectable difference in reader dissociation from chromosomes as cells entered mitosis in the absence of H3T3ph in this study. It was proposed instead that there was more rapid re-recruitment of Dido3 and ING1 to chromosomes in telophase, but the difficulty in distinguishing different stages of late mitosis using fixed cell images, and the use of a Haspin inhibitor whose selectivity is poorly characterized[88], means that care must be taken in interpreting these qualitative results. By contrast, we find no change in the loss and re-recruitment of TFIID, Dido3, and ING2 to chromosomes during mitosis using live and fixed cell imaging, either in cells genetically lacking Haspin, or in cells where Haspin was inhibited by the well-

characterized inhibitor 5-iodotubercidin, in which H3T3ph is undetectable[59,67,68]. Instead, increased chromosome condensation might influence reader protein access to chromatin in mitosis or, perhaps more likely, direct phosphorylation of transcription factors may be required for mitotic displacement (Fig. 8). Indeed, evidence for such a mechanism involving Cdk1-Cyclin B has been found for a number of factors including TFIID[14,51,89], and differential regulation of such phosphorylation could allow locus-specific control of displacement.

The technical difficulties inherent in determining if crosstalk between histone marks occurs in cells are well-known[61,62,90]. Our study provides an example of an effective set of approaches to this general problem. Importantly, we used CIDOP-seq[54] to confirm that the PHD finger of TAF3 is able to bind to sites containing H3K4me2/3 in both mitotic and interphase chromatin in vitro, even in regions near centromeres that have the highest overall levels of H3T3ph in mitosis. This approach may be useful for analyzing the function of other combinatorial histone marks without the uncertainties introduced by the use of antibodies that inevitably have incompletely defined specificity for naturally modified histones.

What might be the significance of excluding H3T3ph from the vicinity of H3K4me3? One possibility is that it allows H3K4me2/3 function and promoter identity to be maintained during mitosis, perhaps including a basal level of mitotic transcription. The absence of H3T3ph would prevent the recruitment and functional impact of H3T3ph-dependent reader proteins at the promoters of active genes in mitosis. For example, Aurora B, which is recruited by H3T3ph as part of the CPC[36,37,39], can alter transcription factor activity[91–94], which might need to be avoided during mitosis. Exclusion of H3T3ph could also permit H3K4me2/3 functions in mitosis, unhindered by competition from adjacent H3T3ph-binding proteins or methyl-phos switching activity. Indeed, although most H3K4me2/3 readers are largely displaced in mitosis, a subset of promoters may retain H3K4me2/3 reading proteins[51,95]. In addition, condensins (which facilitate chromosome compaction in mitosis) appear to be loaded onto chromatin at active promoters[96–100] and, for Condensin II, this may be facilitated by recognition of H3K4me3[101]. Also, notably, controlling release at the level of the reader protein (e.g. by phosphorylation of the reader), rather than at the level of chromatin (e.g. by H3T3ph), allows the possibility of selective release of some but not other reader proteins.

Our results leave open the possibility that some readers of unmethylated or monomethylated H3K4 might still be regulated by H3T3ph. Other work has suggested that mutation of the H3K4me0 reader DNMT3A, to render it insensitive to H3T3ph, increases its association with H3T3ph-containing regions of mitotic chromosomes[57]. These results provide initial evidence to support the importance of H3T3ph in regulating the localization of H3K4me0 readers in cells. Another H3K4me0 reader, PHF21A/BHC80 is strongly displaced from H3 peptides by H3T3ph in vitro[49,73]. However, we were unable to detect any effect of Haspin knockout on the displacement of the PHF21A/BHC80-containing LSD1 complex from mitotic chromosomes. Our findings highlight that, regardless of the results of in vitro binding assays, the effect of combinations of histone marks on the binding of histone reading proteins must be examined in cells on a case-by-case basis before conclusions can be drawn about functional significance. A H3T3-H3K4me2/3 methyl-phos switch can be created in vitro, but our results suggest this mechanism does not function broadly in vivo and is unlikely to be involved in widespread transcriptional repression in cells.

## Methods
### Antibodies
The rabbit monoclonal antibody and dilutions used for immunofluorescence microscopy (IF), immunoblotting (IB), ELISA, ChIP, and flow cytometry were: Histone H3K4me3 (C42D8, Cell Signaling Technology #9751, RRID:AB_2616028, 1:500 for IB, 1:100,000 for ELISA, 3 μg/sample for ChIP). The rabbit polyclonal antibodies used were: Histone H3T3ph (B8634, 1:1000 for IF, 1:500 for IB, 1:4000 for ELISA, 8 μg/sample for ChIP)[35]; Histone H3 (Abcam ab1791, RRID:AB_302613, 1:100,000 for IB); Dido/DATF1 (Invitrogen PA5-101330, RRID AB_2850767, 1:200 for IF); ING2 (Sigma HPA019486, RRID:AB_1851780, 1:500 for IF); LSD1/KDM1A (Abcam ab17721, RRID:AB_443964, 1:200 for IF); GFP (Invitrogen, A-11122, RRID:AB_221569, 1:200 for IF). The mouse monoclonal antibodies used were: Histone H3T3ph (16B2, 1:500 for IF)[37]; TAF3 (39TA-2F5; Invitrogen MA3-074, RRID:AB_2633320, 1:250 for IF); Cy5-conjugated MPM2 antibody (Sigma 16-220, RRID:AB_442398, 1:100 for flow cytometry); GST (GST 3-4 C, Invitrogen 13-6700, RRID:AB_2533028, 1:5000 for IB and IF). The chicken antibody and dilution used was: GFP (Abcam ab13970, RRID:AB_300798, 1:1000 for IF). The guinea pig antibody and dilution used was: CENP-C (MBL PD030, RRID:AB_10693556, 1:2000 for IF).

Donkey secondary antibodies used were: anti-rabbit IgG Alexa Fluor Plus 594 (Invitrogen A-32754, RRID:AB_2762827, 1:1500 for IF); anti-mouse-IgG Alexa Fluor 488 (Invitrogen A-21202, RRID:AB_141607, 1:1500 for IF); anti-mouse IgG Alexa Fluor Plus 594 (Invitrogen A-32744, RRID:AB_2762826, 1:1500 for IF). Goat secondary antibodies used were: anti-rabbit-IgG Alexa Plus Fluor488 (Invitrogen A-32790, RRID:AB_2762833, 1:1500 for IF); anti-chicken IgG Alexa Fluor 488 (Invitrogen A-11039, RRID:AB_142924, 1:1500 for IF); anti-guinea pig IgG Alexa Fluor 647 (Invitrogen A-21450, RRID:AB_2735091, 1:1500 for IF); anti-rabbit IgG-HRP (Cell Signaling Technology #7074, RRID:AB_2099233, 1:10,000 for IB, 1:2000 for ELISA). Horse secondary antibody used was anti-mouse IgG-HRP (Cell Signaling Technology #7076, RRID:AB_330924, 1:10,000 for IB, 1:2000 for ELISA).

### Peptides
The peptides used were:

H3(1-21) unmodified (ARTKQTARKSTGGKAPRKQLA-GGK-biotin, Abgent (custom)),

H3T3ph (ARTphKQTARKSTGGKAPRKQLA-GGK-biotin, Abgent (custom)),

H3T3ph (ARTphKQTARKSTGGKAPRKQL-K-biotin, Epicypher #12-0072),

H3T3phK4me1 (ARTphKme1QTARKSTGGKAPRKQLA-GGK-biotin, Severn Biotech (custom)),

H3T3phK4me2 (ARTphKme2QTARKSTGGKAPRKQLA-GGK-biotin, Severn Biotech (custom)),

H3T3phK4me3 (ARTphKme3QTARKSTGGKAPRKQLA-GGK-biotin, Abgent (custom)),

H3T3phK4me3 (ARTphKme3QTARKSTGGKAPRKQL-K-biotin, Epicypher #12-0069),

H3K4me1 (ARTKme1QTARKSTGGKAPRKQLA-GGK-biotin, Millipore #12-563),

H3K4me2 (ARTKme2QTARKSTGGKAPRKQLA-GGK-biotin, Millipore #12-460),

H3K4me3 (ARTKme3QTARKSTGGKAPRKQLA-GGK-biotin, Millipore #12-563),

H3K4me3K9ac (ARTKme3QTARKacSTGGKAPRKQLA-Ahx-biotin, Alta Histone set 3 library (B1)),

H3K4me3K9me3 (ARTKme3QTARKme3STGGKAPRKQLA-Ahx-biotin, Alta Histone set 3 library (B2)),

H3K9me1 (ARTKQTARKme1STGGKAPRKQLA-GGK-biotin, Millipore #12-569),

H3K9me2 (ARTKQTARKme2STGGKAPRKQLA-GGK-biotin, Millipore #12-430),

H3K9me3 (ARTKQTARKme3STGGKAPRKQLA-GGK-biotin, Millipore #12-568),

H3S10ph (ARTKQTARKSphTGGKAPRKQLA-GGK-biotin, Millipore #12-427),

H3T11ph (ARTKQTARKSTphGGKAPRKQLA-GGK-biotin, Abgent (custom)),

H3K14me3 (Ac-RKSTGGKme3APRKQLATKAARKS-Ahx-biotin, Alta Histone set 3 library (D1)),

H3(15-36) unmodified (APRKQLATKAARKSAPATGGVK-GGK-biotin, Anaspec #AS-65415),

H3T22ph (APRKQLATphKAARKSAPATGGVK-GGK-biotin, Eurogentec (custom)),

H3K23me3 (APRKQLATKme3AARKSAPATGGVK-GGK-biotin, Eurogentec (custom)),

H3K27me3 (Ac-LATKAARKme3SAPATGGVKKPHR-Ahx-biotin, Alta Histone set 3 library (F4)).

## Cell culture and transfection

HeLa S3 (ATCC CCL-2.2; RRID:CVCL_0058), parental and Haspin knockout D2 Hela cells[68], human GFP-TAF5 U2OS cells[49], and doxycycline-inducible mouse GFP-TAF3 HeLa FRT[47] and human GFP-TAF5 HeLa FRT cells[102] were maintained in high glucose DMEM with 10% (v/v) FBS, 100 U/ml penicillin and streptomycin and 2 mM L-glutamine at 37 °C and 5% $CO_2$ in a humidified incubator. Doxycycline inducible lines were induced for 24 h with $0.5 − 1\,\mu g/ml$ doxycycline. Where stated, cells were treated with $1\,\mu M$ 5-iodotubercidin for 2 h. For RNAi experiments, GFP-TAF5 U2OS cells were transfected with 100 nM Haspin siRNA (5′-GGCAUCUGAUCCUGAAAGTT-3′ from Integrated DNA Technologies) or *Silencer* negative control #1 siRNA (AM4611; ThermoFisher) using Lipofectamine RNAiMAX (Thermo-Fisher) and analyzed after 48 h.

Parental and Haspin knockout HeLa cells stably expressing GFP-TAF5 were obtained by cotransfection with pcDNA5/FRT/TO/GFP-TAF5[102] and a pPGKpuro resistance plasmid (in a 9:1 ratio) using Xtreme Gene 9 (Roche). After 48 h, $2\,\mu g/ml$ puromycin was added and, after an additional 10 days, cells expressing moderate levels of GFP-TAF5 were isolated by fluorescence-activated cell sorting.

## Cell synchronization

For ChIP-seq experiments, mitotic HeLa cells were removed from cultures at approximately 25% confluency by shake off before adding 2.5 mM thymidine to the remaining adherent cells for 18 h. Thymidine was removed for 8 h, and re-added for 18 h before an 8 h release into medium lacking thymidine. Nocodazole was added at $0.5\,\mu M$ for 5 h and mitotic cells collected by gentle shake off.

To assess synchronization by flow cytometry, cells were fixed in ice-cold 70% ethanol and permeabilized with 0.1% Triton-X100 in PBS for 5 min. After centrifugation, cells were resuspended in 0.1% BSA, PBS with 1/100 Cy5-conjugated MPM2 antibody for 1 h. Cells were then washed and resuspended in PI staining buffer (50 µg/ml propidium iodide and 0.2 mg/ml RNase A in PBS) and analysed on a FACSCanto II cell analyser (BD Biosciences) and FCS Express 7 (De Novo Software).

## Recombinant proteins and peptides

We used recombinant biotinylated human mononucleosomes (Epicypher 16-0006), H3K4me3 mononucleosomes (Epicypher 16-0316), and H3T3ph and H3T3phK4me3 mononucleosomes (Epicypher, custom). Recombinant human 6His-Haspin residues 471–798 was produced as previously described[35].

Plasmids encoding recombinant WT (RRID:Addgene_92100) and M880A mutant human GST-TAF3-PHD were kindly provided by Albert Jeltsch[54]. *E. coli* BL21 cells carrying each plasmid were grown in LB medium at 37 °C in the presence of $100\,\mu M$ $ZnSO_4$ to an $A_{600}$ of 0.6, then induced with 1 mM IPTG overnight at 20 °C. Collected cells were resuspended in 20 mM HEPES, 500 mM KCl, 0.2 mM DTT, 1 mM EDTA, 10% glycerol, 1 mg/ml lysozyme, 1% Benzonase and cOmplete Protease Inhibitor Cocktail (Roche), pH 7.5, and disrupted by probe sonication. GST-TAF3 PHD proteins were purified using the Pierce GST Spin Purification Kit (ThermoFisher) and dialyzed against 20 mM HEPES,

200 mM KCl, 0.2 mM DTT, pH 7.5 with 10% glycerol for 2 h, then with 60% glycerol overnight.

## ELISAs and protein interaction assays

For characterization of antibody binding to histone peptides, Streptavidin Coated High Capacity 96 well plates (Pierce) were washed 3 times with TBST (TBS, 0.05% Tween20) before adding 100 µl $0.1\,\mu M$ peptide in TBST. After incubation at room temperature for $1 − 2$ h, wells were washed and 100 µl primary antibody in TBST, 0.1% BSA were added and incubated for $1 − 2$ h. After washing, 100 µl 1/2000 HRP-conjugated anti-rabbit or anti-mouse IgG secondary antibodies (Cell Signaling Technology) were added for 1 h at room temperature before washing again. The signal was detected using TMB substrate (Pierce) according to the manufacturer's instructions. Absorbance was measured at 450 nm (signal) and 570 nm (background) using a Polarstar Omega microplate reader (BMG Labtech). ELISAs using recombinant mononucleosomes were carried out in 384-well high-capacity streptavidin-coated plates (Pierce). Mononucleosomes were used at $0.1\,\mu M$ in 40 µl TBST. Antibodies were used as above at 40 µl per well. For antibody binding assays in the presence of sheared chromatin, 0.8 µg H3T3ph antibody B8634 and $0.1\,\mu M$ mononucleosomes were incubated with 20 µl sheared chromatin (obtained as for ChIP-seq), shaking at 300 rpm, for 2 h in a 384-well plate. Samples were then moved to a 384-well high capacity streptavidin-coated plate and antibody binding was detected by ELISA as just described.

Analysis of GST-TAF3-PHD binding to histone peptides (coated at $0.1\,\mu M$ in 40 µl TBST) was carried out in 384-well high capacity streptavidin-coated plates (Pierce). GST-TAF3-PHD (WT or M880A mutant) were made to equivalent concentrations (~0.1 mg/ml) in interaction buffer (20 mM HEPES, 300 mM KCl, 0.1 mM DTT, 10% glycerol, pH 7.5). After GST-TAF3-PHD binding to peptides for 1 h at room temperature, the plates were washed 3 times with interaction buffer, then once with TBST, before anti-GST antibody (Invitrogen) binding and detection as described above.

## In vitro kinase assays

Reactions contained recombinant 6His-Haspin, $0.1\,\mu M$ recombinant biotinylated human mononucleosomes, and 0.2 mM ATP in K buffer (50 mM Tris, 10 mM $MgCl_2$, 1 mM EGTA, 10 mM NaF, 20 mM β-glycerophosphate, 1 mM PMSF, pH 7.5 with PhosSTOP (Merck)), and were carried out in 384-well microplates at 37 °C for 30 min. Samples were then moved to a 384-well high-capacity streptavidin-coated plate, and phosphorylation detected using H3T3ph antibody (B8643) as described for ELISA.

For kinase assays using whole cell extract, HeLa cells were treated with 300 nM nocodazole for 15 h, collected by shake off, and lysed at 4 °C in 50 mM Tris, 0.25 M NaCl, 0.1% Triton X100, 10 mM $MgCl_2$, 2 mM EDTA, 1 mM DTT, pH 7.5, with protease inhibitor cocktail (Sigma P8340), PhosSTOP, 1 mM PMSF, 0.1 µM okadaic acid, 10 mM NaF and 10 mM β-glycerophosphate, at approximately $30 × 10^6$ cells/ml. Extracts were immediately flash frozen in liquid nitrogen. Kinase reactions contained 0.35 µM biotinylated peptide, 0.2 mM ATP, and up to 5% mitotic extract in K buffer. Where indicated, 5-iodotubercidin was included at 10 µM. Phosphorylation was detected as described above.

## Immunofluorescence microscopy

Cells grown on glass coverslips or 8-well chamber slides (Eppendorf) were either fixed for 10 min with 4% (v/v) formaldehyde in PBS, washed twice with PBS, and permeabilised for 2 to 5 min with 0.1% Triton X100 in PBS, or were treated with ice-cold methanol for 10 min. After washing twice with PBS, samples were blocked for 1 h with 1 to 5% BSA in PBS, 0.05% Tween 20 (PBST). Primary antibodies in blocking buffer were added for $1 − 2$ h at 37 °C then washed twice with PBST before incubating with fluorophore-conjugated secondary antibodies for

approximately 1 h in blocking buffer. After washing twice with PBST, once with PBS, and once with $H_2O$, mounting was carried out with ProLong Glass with NucBlue (ThermoFisher).

Confocal imaging was performed using a Leica SP8 confocal microscope with a 63x NA1.4 Plan Apo CS2 Oil objective using Leica LasX v3 software, or a Nikon A1 confocal microscope equipped with a 60x NA1.4 Apo Oil λS DIC N2 objective using Nikon Elements 5.22 software, or a Visitech VT-iSIM super-resolution Nikon TiE-based microscope (Visitech, UK) with instant SIM scanhead coupled to two Hamamatsu Flash4 v2 cameras (Hamamatsu, Japan) via a dual port splitter, using a 60x NA1.4 Plan Apo VC Oil DIC N2 objective and Elements 5.21.03 software (Nikon, Japan). Images were processed in ImageJ 2.3.0[103] and Adobe Photoshop 2022, and are displayed as maximum intensity projections of the central 5 slices.

### Live imaging
Live imaging was performed in glass-bottomed 35 mm FluoroDishes (WPI) in FluoroBrite DMEM medium (ThermoFisher). DNA was stained with 250 nM SiR-DNA (Spirochrome). Multiple regions of cells were imaged through ten 2.5 μm sections at 5 min intervals for 16 h using the Visitech VT-iSIM super-resolution microscope described above, with a 40x NA1.3 S Fluor Oil objective and Elements 5.21.03 software (Nikon, Japan). A humidified environment at 37 °C and 5% $CO_2$ was maintained with an Okolab whole microscope and stage-top incubator (Okolab, Italy).

### Image quantification
GFP-TAF5 to SiR-DNA intensity ratios on chromatin in live cell imaging experiments were quantified using ImageJ 2.3.0. Using sum intensity projections, automated Otsu thresholding on the DNA channel defined masks containing the chromatin regions within dividing cells. Watershedding and manual deletion of chromatin regions from adjacent (non-dividing) cells was then carried out. The intensities of the DNA and GFP signals within the defined regions were then recorded for each time point. Cell-specific mean interphase cytosolic background signals for SiR-DNA and GFP channels were subtracted from all corresponding data points. The results were standardized such that the GFP-TAF5/SiR-DNA intensity ratio at the time point immediately preceding nuclear envelope breakdown equaled 1. The first time point at which initial separation of two chromosome masses was evident was taken as anaphase onset. TAF3 and H3T3ph intensity ratios on chromatin in immunofluorescence experiments were quantified using ImageJ 2.9.0. Using sum intensity projections, automated Otsu thresholding on the DNA channel defined masks containing the chromatin regions of cells. The intensities of the DNA, TAF3, and H3T3ph signals within the defined regions were then recorded for each cell. Mean cytosolic background signals for each corresponding channel were subtracted. The results were standardized such that the mean TAF3/DNA intensity ratio of interphase nuclei, and the H3T3ph/DNA intensity ratio of mitotic (i.e. prometaphase and metaphase) chromatin, each equaled 1.

### Chromatin isolation and shearing
Cells were fixed in medium with 1% formaldehyde for 10 min at room temperature before quenching with 125 mM glycine for 5 min. Cells were washed once with PBS before adding cold lysis buffer iL1b (from the Auto iDeal ChIP-seq Kit for Transcription Factors, Diagenode C01010058). Asynchronously growing cells (collected by scraping) and mitotic-enriched cells (as described earlier) were suspended at approximately 50 ×$10^6$ in 50 ml in buffer iL1b and incubated at 4 °C on a roller for 20 min, pelleted, and resuspended in 30 ml buffer iL2 for 10 min at 4 °C. Pelleted material was then resuspended in buffer iS1b with protease inhibitors (100 μl per 3 million cells) and incubated on ice for 10 min. The sample was divided into 270 μl aliquots in Pico sonication tubes (Diagenode C30010016) and sheared using a

Bioruptor Pico (Diagenode). Asynchronous samples were sheared for 8 cycles of 15 s with 30 s breaks, and mitotic samples for 5 cycles of 30 s, to yield DNA fragments of 100−500 bp. Samples were then centrifuged at 16,000 g for 10 min and the supernatant collected and pooled. Where phosphatase inhibitors were used, PhosSTOP (Roche) was added to buffers iL1b, iL2, and iS1b. Sheared chromatin in aliquots of 210 μl (at approximately 1.5 mg protein/ml based on NanoDrop (ThermoFisher) quantification at $A_{280}$) was snap frozen in liquid nitrogen and stored at −80 °C.

### SDS-PAGE and immunoblotting
Sheared chromatin was diluted in NuPAGE LDS sample buffer (ThermoFisher) with 10% DTT and 1% Benzonase and incubated at 95 °C for 15 min. For analysis of precipitated proteins in CIDOP experiments, elution from washed MagneGST beads was carried out by boiling in NuPAGE LDS buffer for 10 min. Samples were electrophoresed on 4 − 12% Bis-Tris gels and analyzed by immunoblotting using standard procedures.

### ChIP-seq
The Auto iDeal ChIP-seq for Transcription Factors protocol (Diagenode) was followed using 200 μl sheared chromatin and the IP-Star Compact Automated System and the direct ChIP iPure 200 method (Diagenode). The antibody coating step was for 3 h, immunoprecipitation for 15 h, and washes for 5 min each at 4 °C. Antibodies were used at 3 − 8 μg and DiaMag protein A-coated magnetic beads between 10 and 25 μl accordingly. As crosslinks were reversed, Proteinase K was also added. After crosslink reversal, RNase A was added for 30 min at 37 °C. DNA was purified from the ChIP samples (and from an aliquot of sheared chromatin as input) using iPure beads (Diagenode) as stated in the Auto iDeal protocol. Finally, samples were quantified using a Qubit 4 fluorometer and the dsDNA HS Assay kit (ThermoFisher). ChIP libraries were prepared using NEBNext Ultra II DNA Library Prep Kit for Illumina per manufacturer's instructions (NEB). Sequencing was carried out on Illumina's NextSeq 500 platform to produce 75 bp single-end reads at a minimum of 50 million reads per sample. ChIP-seq was carried out in 2 (for input and H3K4me2/3) or 3 (for input and H3T3ph) biological replicates.

### CIDOP-seq
Sheared chromatin (100 μl aliquots) was supplemented with 95 μl 0.1% BSA, 50 mM Tris, 434 mM NaCl, 0.5% NP-40, 2 mM DTT, 1 mM $ZnCl_2$, pH 7.4, with protease inhibitors (Diagenode). Chromatin was then precleared with MagneGST magnetic beads (Promega) for 1 h at 4 °C, and then incubated with approximately 75 μg wild type or M880A recombinant GST-TAF3-PHD overnight at 4 °C. MagneGST beads were washed with PB300 (50 mM Tris pH 7.4, 300 mM NaCl, 0.5% NP-40, 2 mM DTT) then added at 12.5 μl per sample for 1 h. The beads were then washed three times for 10 min each at 4 °C in buffer PB300, once in 10 mM Tris, pH 7.4, then resuspended in 100 μl buffer iE1 (Diagenode) and incubated rotating for 30 min at room temperature before removing the beads. For the input, 2 μl of precleared chromatin was added to 98 μl buffer iE1 (Diagenode). To samples and inputs, 4 μl iE2 was added then Proteinase K was added for 4 hours at 65 °C before adding RNase A for 30 min at 37 °C. DNA was purified as for ChIP-seq samples following the Diagenode iPure protocol using reagents from the Auto iDeal ChIP-seq kit (Diagenode). DNA quantification, library construction and sequencing were performed as for ChIP-seq. Two (asynchronous) or three (mitotic) biological replicates were produced for each condition (input, GST-TAF3-PHD WT, GST-TAF3-PHD M880A).

### Sequencing data formatting and visualization
FastQC v0.11.7 (http://www.bioinformatics.babraham.ac.uk/projects/fastqc/) and MultiQC v1.7216 were used to assess the quality of

sequencing reads. Reads were aligned to reference genome GRCh38.p12 (GCA_000001405.27) using Bowtie2 v2.3.4.2[104] with the --very-sensitive option. Samtools v1.9[105] was used to create a bam file of the alignments in addition to sorting and indexing. Bam files were converted to bigwigs using bedtools v2.29.2 genomeCoverageBed[106] followed by bedSort and bedGraphToBigwig (UCSC Genome Browser 'kent' bioinformatic utilities; http://genome.ucsc.edu/) for data visualization in IGV v2.4.16[107]. Ideograms were created in R v4.2.1 using the package KaryoploteR v1.22.0[108] and the density of ChIP-seq reads in bedgraph format were plotted.

ChIP-seq replicates were merged using Sambamba v0.7.1[109] and normalized to the appropriate inputs using deepTools2 v3.5.1 bamCompare[110], scaling by read depth and reporting the output as a ratio in bigwig format.

Peak calling of H3K4me2/3 in ChIP-seq data was carried out using MACS2 v2.1.1.20160309[111] with the --broad option and standard parameters. Peaks called in more than one replicate were retained using bedtools v2.29.2 intersect, sort, and merge tools plus bedops v2.4.39[112] using option --everything.

### Sequencing data analysis

Metagene plots and heatmaps were plotted from input-normalized bigwigs using deepTools2 v3.5.1 computeMatrix with the reference-point option, followed by either plotProfile or plotHeatmap with standard parameters. TSSs were from the UCSC Table Browser and H3K4me2/3 peaks called as described above.

To define the H3T3ph enriched centromere proximal regions, we used a binarisation approach because standard peak calling algorithms perform poorly on broad domains such as centromeric H3T3ph. Bam files were binarized using ChromHMM v1.20 Binar-izeBam with the -center option[113] with a bin size of 200 bp to determine H3T3ph signal per bin. The input for each sample was used as the control data set. Binarised H3T3ph files of all three replicates were then merged and analysed in R v4.2.2. The largest contiguous block of H3T3ph positive bins on each chromosome was identified (which corresponded to the centromere in each case). Then, to define the two ends of the entire H3T3ph domain on each chromosome, we extended outwards from the contiguous block. The extension continued until 3000 consecutive bins (600 kb) showing no H3T3ph signal were reached. Using this method, 95.2% of the H3T3ph positive bins across all chromosomes were included in the centromere proximal regions. The TSSs within this region were then identified and used in the analysis of centromere-proximal regions in CIDOP-seq data. To complement this analysis, we performed the same procedure for TSSs outside this region (i.e. chromosome arms).

To assess if H3T3ph was present at a subset of promoters with H3K4me2/3, we collected all 200 bp ChromHMM binarization bins that contained both H3T3ph and H3K4me2/3 in all mitotic replicates. If any of these double positive bins were found in a region ± 1 kb of a TSS, then that promoter was deemed to have both H3T3ph and H3K4me2/3 (0.3% of all TSSs with H3K4me2/3). Possible over-representation of the corresponding genes in particular cytogenetic locations, gene ontology groups, or KEGG pathways was then analyzed using WebGestalt 2019[114]. Possible over-representation of promoters with H3T3ph and H3K4me2/3 among genes whose nascent transcripts were identified as enriched or depleted in mitotic chromatin compared to asynchronous chromatin by Liang et al. [15] was assessed by gene list overlap analysis followed by $\chi^2$ tests.

The Pearson correlation of CIDOP-seq or ChIP-seq samples was calculated using deepTools2 v3.5.1 multiBigwigSummary bins and input-normalized bigwigs, with standard parameters, followed by plotCorrelation --corMethod pearson, with standard parameters, to produce a heatmap.

### Statistical analyses

Statistical analyses for binding and kinase assays were carried out in Prism 9.4.1 (GraphPad) with non-normalized data using one- or two-way repeated measures ANOVA or a mixed effects model, followed by Dunnett's or Šídák's multiple comparisons tests, as indicated in figure legends. Adjusted $p$ values are reported. Statistical analysis for live cell imaging was carried out in Prism 9.4.1 using a two-way mixed effects model ANOVA. Statistical analyses for immuno-fluorescence experiments were carried out in Prism 10.0.0 with non-normalized data using Student's $t$ test. Pearson $\chi^2$ tests for gene enrichment analyses were conducted in Excel 16.16.26 (Microsoft) with $\alpha = 0.05$.

### Reporting summary

Further information on research design is available in the Nature Portfolio Reporting Summary linked to this article.

## Data availability

Raw and processed ChIP-seq and CIDOP-seq data generated in this study have been deposited in the public Gene Expression Omnibus under GEO accession code GSE226768. The human reference genome GRCh38.p12 (GCA_000001405.27) used in this study is available from the National Center for Biotechnology Information [https://www.ncbi.nlm.nih.gov/datasets/genome/GCF_000001405.27/], and the HeLa H3K4me3 data used in Supplementary Fig. 3 is available from the ENCODE Portal [https://www.encodeproject.org/files/ENCFF489CIY/]. Source data are provided with this paper.

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

## Acknowledgements

We would like to sincerely thank the late Dr Robert Stones, of the Newcastle University Bioinformatics Support Unit, for his contributions to the early stages of this work. We also thank Dr Albert Jeltsch (Institute of Biochemistry and Technical Biochemistry, University of Stuttgart) for providing constructs encoding the TAF3 PHD finger, Dr Marc Timmers (German Cancer Research Center, University of Freiburg) for constructs and cell lines expressing GFP-TAF3 and TAF5, Dr Hiroshi Kimura for H3T3ph antibody 16B2 (Tokyo Institute of Technology), and Dr Fangwei Wang (Life Sciences Institute, Zhejiang University) for Haspin knockout cells. The authors gratefully acknowledge the Newcastle University BioImaging Unit, Genomics Core Facility, Bioinformatics Support Unit, and Flow Cytometry Core Facility (FCCF) for their support and assistance in this work. This study was funded by a Wellcome Trust Investigator Award (106951/Z/15/Z) and a Royal Society Wolfson Research Merit Award (WM130089) to J.M.G.H., an EPSRC DTP (Biological Informatics) PhD Studentship to M.H., a Barbour Foundation PhD Studentship to J.L.M., and by a J.G.W. Patterson Foundation grant to L.G.

## Author contributions

R.J.H.: conceptualization, investigation, formal analysis, visualization, and writing - original draft. M.H.: software and formal analysis. M.D.L.: investigation. T.N.C.: investigation. B.W.: investigation. J.L.M.: formal analysis. J.M.C.: investigation and methodology. L.G.: methodology. L.P.: investigation and methodology. D.R.: conceptualization, writing - review & editing, supervision, and funding acquisition. J.M.G.H.: conceptualization, formal analysis, visualization, writing - original draft, supervision, project administration, and funding acquisition.

## Competing interests

The authors declare no competing interests.
