## [Peer Review File · Nature Communications]

Release of Histone H3K4-reading transcription factors from chromosomes in mitosis is independent of adjacent H3 phosphorylationREVIEWER COMMENTS

Reviewer #1 (Remarks to the Author):

Please read review in the attached file, as it contains links to relevant publications.

Release of Histone H3K4-reading transcription factors from chromosomes in mitosis is independent of adjacent H3 phosphorylation, by Harris *et al.*

Summary of the main findings of the study.

The study provides compelling evidence that the methyl-phos switch mechanism does not function *in vivo* and is unlikely to be involved in widespread transcriptional repression in cells during mitosis. Therefore, this work indicates that H3T3ph is implausible to be responsible for transcriptional downregulation during cell division. For this reason, this work presents significant information for the field of mitosis and by extension for general cell biology research shedding new light on the mechanisms that regulate gene expression in human cells.

Strengths and limitations of this work.

The conclusions that this work presents are relevant for enhancing our knowledge about gene expression regulation. The main strength of this work lies in the comprehensive use of multiple *in vivo* and *in vitro* methodologies such as live imaging and IF, ELISAs on chemically synthesized peptides, Chip-seq, CIDOP-seq, Kinase assays, and genome-wide distribution analysis of H3T3ph and H3K4me3. The work is furthermore complemented by different approaches to study H3T3ph and H3K4me3 related proteins: Haspin kinase (responsible for H3T3 phosphorylation) and several H3K3-reading proteins (TAF3, TAF5, ING2, Dido3 and ING2).

The conceptualization of this work is impressively thorough, incorporating a wide range of techniques and approaches to address the research question at hand. The results are neatly presented and properly discussed. Although no clear limitations were identified, some suggestions for improvement are provided below.

Comments and suggestions:

Introduction:

Please indicate which histone marks are critical for the sentence mentioned in lines 39-41 page 3 in relation to BRD4 bookmarking (references 32, 41, 42).

Suggestion: In page 4 lines 6-7 authors might wish to cite some of the recent reports about Haspin activity during meiosis, as its role in H3T3 phosphorylation is conserved in both types of mammalian cell division (mitosis and meiosis). Examples of Haspin implications in meiosis and their relationship with histone modifications during meiosis are DOI: [10.1242/jcs.259546](https://doi.org/10.1242/jcs.259546), DOI: [10.1091/mbc.E16-12-0850](https://doi.org/10.1091/mbc.E16-12-0850), DOI: [10.1242/jcs.158840](https://doi.org/10.1242/jcs.158840) or DOI: [10.1242/jcs.189340](https://doi.org/10.1242/jcs.189340).

Results:

This reviewer has some specific major and minor suggestions, which are explained below in the same order of appearance of the results section:

- **Page 5 Lines 18-21**

In Figure 1 panel B, why T3ph is not included? (major)

Equally to panel A, where anti-H3T3ph antibody B8634 is used against peptides carrying H3K4 methylation, panel B should indicate if anti-H3K4m3 antibody C42D8 detects peptides carrying H3T3ph alone. The graph includes H3K4me1, K4me2 or K4me3 in addition to H3T3ph, but it does not include H3T3ph alone.

- **Page 5 line 45 (minor)**

For Extended Data/ED-Figure 2C, the text reads that “we also confirmed that the level of H3T3ph was largely preserved in preparations of sonicated chromatin from mitotic cells compared to whole HeLa cell lysates”, however all the figures and the material and methods section refers to this as “sheared chromatin”. This reviewer is aware that the material methods sections explains that shearing protocol is done by sonication (lines 32-33 page 16) but please unify and mention the same terminology in lines 45-46 of pag 5.

- **Page 6 line 28 (minor)**

For Figure 3, the text reads “*the detected H3T3 kinase activity was strongly inhibited by a Haspin inhibitor*”. As this is the first time that the inhibitor is mentioned in the text, please include here the specific information: indicate that inhibition is done with 5-iodotubercidin, in which H3T3ph is undetectable (cite references 57,65,66). This information is included for the first time in further pages of the original manuscript (in page 10). Please also indicate here in page 6 the concentration of use, and specify if it is the same one used for previous works of the group (reference 57).

Regarding the concentration of 5-Iodo inhibitor, the information provided in the manuscript is not clear. In page 16 line 7, it says that “*cells were treated with 1 μ M 5-iodotubercidin for 2 h*”. In contrast, page 17 line 20 says that “*Where indicated, 5-iodotubercidin was included at 10 μ M.*”. Please clarify the concentration used for all the experiments in this work: 1 μ M or 10 μ M.

- **Page 6 line 37 (recommendation)**

This reviewer believes that the schematic figure cited in line 28/page 6 (red box of ED-Figure 4) is useful and helps the understanding the relevance of the study. As a recommendation, a similar image, or a more detailed version summarizing the main conclusions of the work, could be included as Figure 8 within the discussion section. Incorporating this figure will surely bolster the overall message conveyed by this publication.

- **Page 7, lines 8-10 (major)**

Why does the Figure 5 not include quantification of endogenous TAF3 in the presence (Wild Type) or absence (Haspin KO) of H3T3ph in HeLa cells?

Live imaging results obtained for GFP-TAF5 in HeLa and U2OS are very robust. However, although the IF images of the endogenous protein are very representative, this reviewer believes that Figure 5 will benefit from a quantification analysis. This could be done by quantitative analysis of the relative signal intensities making sure that the acquisition time was fixed for all acquired images, and the quantification was only performed using the original unmodified images. Among other approaches, measuring the integrated fluorescence density in individual nuclei using ImageJ by creating a binary mask with the DAPI staining in a minimum of 10 cells per stage, could be an efficient approach, but author can choose another if they prefer.

Secondly, as far as this reviewer understands, all the experiments of this work are done in HeLa cells, including GFP-TAF5 live images for WT and Haspin KO HeLa cells. However, to study TAF5 under H3T3ph absence, Haspin inhibition with 5-Iodo has been done both in HeLa and also in U2OS cells. Please, explain why TAF3 is the only protein addressed in both cell lines, in case there is a particular reason for it. In this regard, I would recommend a change in order of the figures for a better organization of the results: images of HeLa cells of ED-Figure 5 should be moved to ED-Figure 4 panel C complementing the other results for HeLa Cells in panels A and B, and consequently, the only experiment done with U2OS cells could be separated and moved to ED-Figure 5.

- **Page 8 lines 2-11 (major)**

Please indicate in the text that Figure 7 and ED-Figure 8 are both presented because they show IF of the same proteins but with different fixations: methanol for Figure 7 and formaldehyde for ED-Figure 8. Why is the reason behind this dual approach? Please explain this in the results section.

Regarding this matter, the outcomes obtained for ING2 and LSD1 vary noticeably depending on whether methanol or formaldehyde was used for fixation. On the one hand, ING2 shows different patterns in WT cells if IF is done after methanol (Figure 7) or formaldehyde (ED-Figure 8) fixation. When fixed with the latter, ING2 shows a clear perichromatin localization in WT cells at telophase, whereas in Haspin KO it remains diffuse at the cytoplasm at that stage (ED-Figure 8). Concerning this issue, it is worth noting that ING proteins have been reported to interact with Lamin A in the nuclear envelope, authors might take this into consideration (as an example Han *et al* 2008 <https://www.nature.com/articles/ncb1792>). On the other hand, Figure 7 (methanol fixation) shows

equal signals for LSD1 in WT and Haspin KO cells. In contrast, ED-Figure 8 (formaldehyde fixation) shows that LSD1 is reincorporated to the perichromatin at telophase in WT, but not in Haspin KO. This might be related to the appointed role for LSD1 in the histone demethylation as a regulatory mechanism linking the chromatin state with the nuclear envelope reorganization at the end of mitosis (Schooley *et al* 2015m DOI: [10.1242/jcs.173013](https://doi.org/10.1242/jcs.173013)). Authors are encouraged to clarify the reasons behind the divergent results obtained through different fixation techniques in HeLa cells at telophase, and if desired, discuss and explore potential reasons for the different outcomes observed for ING2 and LSD1 in WT versus Haspin KO (ED-Figure 8).

Discussion: The discussion encompasses all pertinent previous studies on the topic and provides a robust rationale for the findings. That being said, I would like to offer a couple of suggestions:

- This reviewer is aware of space and reference number limitation, but some additional discussion about the transcription at centromeric regions will benefit results obtained from Figure 6B. This could be included in relation to lines 10-16 in page 10. As an example, reviews from Hong Liu (2016) doi: [10.1080/21541264.2015.1127315](https://doi.org/10.1080/21541264.2015.1127315) , or Chan and Wong (2012) DOI: [10.1093/nar/gks921](https://doi.org/10.1093/nar/gks921) are interesting to revise for original references. More specifically, previous works suggesting that the centromeric transcription helps maintain centromeric proteins at centromeres and about the essential role of RNA Polymerase II at the centromeres during mitosis are relevant (Chan *et al* 2012, DOI: [10.1073/pnas.1108705109](https://doi.org/10.1073/pnas.1108705109), Hall *et al* 2012 DOI: [10.1007/s10577-012-9297-9](https://doi.org/10.1007/s10577-012-9297-9)).
- As previously commented for Figure 7 and ED-Figure 8, discussion about the role of LSD1 in nuclear reassembly could also be interesting. In this regard, as mentioned above, previous results indicated that demethylation of histone H3 by LSD1 is involved in the regulation of nuclear reassembly at telophase (Schooley *et al* 2015, DOI: <https://doi.org/10.1242/jcs.173013>).

Reviewer #2 (Remarks to the Author):

The manuscript from Harris et al investigates the interplay of histone H3T3ph and histone H3K4me3 modifications during mitosis. The work offers a reconsideration of previously established findings, where H3T3ph was found to block the binding of reader proteins to H3K4me3 in vitro and the H3T3ph kinase Haspin was shown to promote eviction of H3K4me3-dependent transcription factors from chromatin in cells. In the current study, the authors build on prior results using histone peptides and demonstrate that H3K4me3 can decrease H3T3 phosphorylation in the context of nucleosomes. Importantly, they used ChIP-seq to map H3K4me2/3 and H3T3ph in mitotic cells and found that H3T3ph is not enriched at H3K4me2/3 peaks at gene promoters. Using CIDOP-seq, they further show that the H4K3me2/3 binding PHD domain of TAF3 can still bind to H3K4me3 in mitotic chromatin, suggesting that other signals evict the TFIID complex from chromosomes in mitosis. In line with this, but in opposition to prior results (Varier et al, EMBO, 2010), they find that Haspin depletion or inhibition does not affect the eviction of TAF3 and other H3K4 readers from mitotic chromosomes. The authors conclude that H3T3ph is not a “switch” in vivo.

The main advance of this paper is the demonstration that H3K4me3 and H3T3ph are not overlapping modifications on mitotic chromatin. Thus, although previous work from different labs have shown that H3T3ph is antagonistic towards H3K4me2/3 readers in vitro, this work seeks to make the point that this is not physiologically relevant to the eviction of these readers during mitosis in vivo. These findings are important and will be relevant to researchers in both the transcription and cell division fields. However, much of the work here is either confirmatory or in direct conflict with previous results and does not provide enough evidence to rule out a role for H3T3ph in the eviction of H3K4 readers, in particular TAF3 of the TFIID complex. To be considered for publication in Nature Communications, my opinion is that further evidence is required to support the conclusion that Haspin and H3T3ph do not play a role in the eviction of H3K4 readers.

1) I recommend that Figures 1 and 3 be moved to the supplementary materials section due to the fact that they are mainly supportive findings or have been previously published in some form. I also recommend that Figures 4 and 5 be merged and some aspects be moved to the supplement because they are essentially covering the same point.

2) One of the main points of the paper is that H3T3ph signals are not enriched at H3K4me2/3 peaks in mitotic cells enriched for prometaphase to metaphase. However, since most of the H3T3ph signal is enriched at repetitive centromeres and the peri-centromeric chromatin due to removal of cohesin from the arms by the prophase pathway (the PDS5 subunit of cohesin helps to enrich Haspin), it is unclear how robust the H3T3ph signals are on bulk chromatin and whether the anti-correlation between H3T3ph and H3K4me2/3 is relevant (the amount of H3T3ph is quite low). According to their results, TAF3 is evicted early on during prophase, and so this ChIP-seq dataset may not reflect an accurate snapshot of the chromatin landscape during prophase where H3T3ph has been reported to be more enriched on chromosome arms. To get an idea of the robustness of the H3T3ph signal, I suggest that the authors perform ChIP-seq on Haspin KO cells. To get a better snapshot of cells in a prophase-like state, the authors can deplete WAPL, which forces cohesin and RNA Polymerase II onto the chromosome arms (Perea-Resa et al, Mol Cell, 2020), and would likely increase Haspin-dependent H3T3ph in these regions. With more robust H3T3ph signals in the context of increased RNA Pol II, I feel that the authors can make a stronger case for their claim that “H3K4me3 hinders adjacent H3T3ph deposition in cells”.

3) It was previously shown that Haspin overexpression in interphase leads to TAF3 and TFIID eviction and abrogation of TFIID transcription in U2OS cells (Varier et al). These results provided strong evidence that outside of mitosis, H3T3ph is sufficient for eviction. In the same paper, it was also shown that knockdown of Haspin by RNAi hampered eviction of the TFIID subunit TAF5 in mitosis. One potential view of these results in light of the current manuscript is that H3T3ph can aid in TFIID

eviction, but it is not the primary signal. The role of H3T3ph in this regard may be cell-type or locus specific, as the prior work was performed in U2OS cells and the current work utilized HeLa cells. To investigate this further, the authors should perform live imaging with GFP-TAF3 in U2OS cells depleted of Haspin. In addition, the authors could utilize the LacO-LacI system to target H3T3ph and H3K4me3 to specific LacO repeats in mitotic U2OS and HeLa cells to determine the interplay between these marks and TAF3 recruitment at a defined locus away from centromeres in different cell types.

4) Please describe how the curves were fit in Figure 3. Also, the curve in Figure 3A does not go to completion for the H3K4me3 nucleosomes. Why is 1 nM haspin chosen as 100%? This is an important point since Haspin can be enriched on chromatin by cohesin, and therefore could exist in locally high concentrations. These local concentrations could overcome the inhibitory effects of H3K4me3, and it would be important to understand the relationship between Haspin concentration and phosphorylation in the presence of H3K4me3. This is relevant to the discussion on page 9, lines 34-42, where the authors cast doubt on prior results that utilized Haspin overexpression in interphase cells.

5) It is important to point out in the manuscript that prior work has shown that H3K4me2 is not very inhibitory towards Haspin-dependent H3T3 phosphorylation. This is relevant because, as the authors nicely show in Figure 1, their antibody recognizes both H3K4me2 and H3K4me3 equally. This is particularly relevant for the sentence in line 38, page 6.

Reviewer #3 (Remarks to the Author):

This study by Harris and colleagues challenges the model by which a methyl-phos switch is at least partially responsible for the release of transcriptional activators from mitotic chromatin, which contributes to transcriptional attenuation at mitosis. The focus of the study is on the methyl-phos switch H3K4me2/3 – T3Ph. Phosphorylation of H3T3 by Haspin prevents the recognition of H3K4me3 by the reader TFIID/TAF3, which in turns reduces transcription.

This model is currently based on in vitro data and the authors investigated it further using a combination of elegant in vitro and in-cell approaches. Interestingly, ChIPseq showed that H3T3ph is not present at promoters featuring H3K4me3 during M phase. At the same time, as expected, TFIID/TAF3 occurrence at these promoters decreased compared to interphase. These data thus exclude the possibility of a methyl-phos switch mechanism to exclude TFIID from mitotic chromatin. Although the authors do not have data pointing at an alternative mechanism, this study is very well conducted, very well balanced between in vitro and in-cell experiments and very well presented and written. The results are clearly displayed and relevant controls included. This study will be of interest to a broad readership. Nevertheless, I listed some points below that should be addressed before publication. I hope that they will be helpful.

Major

1) The decreased presence of TFIID/TAF3 at mitotic chromatin vs interphase chromatin is not obvious in Figure 6 and Ext. Figure 7. Especially when looking at heat maps and genome browser snapshots. The authors should adjust the scales of the density plots so that we could clearly see that TFIID presence at mitotic chromatin is decreased compared to interphase (same scale adjustments apply for H3K4me3 metagenes in Figure 2B). Is the amplitude of this decrease uniform genome-wide or stronger at specific subsets of genes. Does it correlate with transcriptional attenuation? (see point 2-4)

2) Genome-wide H3K4me3 ChIP intensity doubles between interphase and M (Figure 2). As for TFIID above, is it a homogeneous increase and does it correlate with gene expression variations?

Although H3K4me3 may increase during M at bivalent genes, I would expect this mark to remain relatively stable from interphase to M in HeLa cells. The increased mitotic presence of H3K4me3 observed by the authors could actually favor retention of TFIID and sustain transcription during M, at least at some genes like housekeeping genes. Or maybe it is only bookmarking and H3K4me3 is not accessible to TFIID due to chromatin compaction. The authors should comment on this and perform transcriptomics (see below).

3) Indeed, gene expression data would benefit this study. Gene expression in interphase and mitosis should be assessed (or collected from a previous study if possible) and integrated with H3K4me3/T3ph ChIPseq and TFIID CIDOP-seq data. These data will bring more depth to current conclusions, which right now seem to draw a uniform model that would apply to most genes and promoters in M phase while previous studies indicated differential regulation of subsets of genes during mitosis (e.g. Palozola et al, Science, 2017).

To illustrate my point, the authors mention that T3ph is "rarely" found at promoter were K4me3 is found. What is the actual proportion of promoters having both T3ph and K4me3? Then what is the status of TFIID at such promoters and what are the expression levels of these genes compared to interphase? Maybe their mitotic downregulation is stronger than at genes were T3ph is not found at the promoter with K4me3?

Together, the points above could strengthen the conclusion that TFIID release from mitotic chromatin and decreased genes expression are not linked to T3ph. They could also reveal subsets of genes at which transcription, histone marks and TFIID recruitment do not correlate fully with the model proposed by the authors, which is likely since ext. fig. 5 indicates that TAF3 substantially remains on mitotic chromatin after Haspin inhibition.

I am just wondering whether, although not predominant in cells, the methyl-phospho switch could still coexist with this new model, at specific subsets of genes, to exclude TFIID and decrease transcription.

Minor

ChIPseq in Fig. 2 and in vitro data in Fig. 3 are clear. But, is K4me3 preventing phosphorylation of T3 during mitosis? To establish a direct link, inhibition of H3K4 methylation could be done, maybe by downregulating the WDR5 protein like in Lauberth et al, Cell, 2013. One would presume that more T3ph will be detected at promoters upon downregulation of K4me3 in ChIPseq or ChIPqPCR experiments.

TFIID binding is not influenced by the distance from the centromere, which suggests that high H3T3ph ChIP intensity around H3K4me3 peaks has no effect on gene expression. Still, I was wondering whether gene expression is differentially regulated between the pericentromeric H3T3ph broad peak, compared to chromosome arms during M phase. This would point at an influence of H3T3ph (or other marks like H3S10ph) on gene expression that is not related to a methyl-phospho switch or regulation of TFIID. The answers to this question could be gathered from results collected in point 3-4 above.

Preventing mitotic condensation of chromosomes in yeast triggers an increased occurrence of H3K4me3 and unscheduled gene expression (Ramos-Alonzo et al, PNAS, 2023). This is linked to Aurora B activity, which can be recruited by Haspin (Hadders et al, J Cell Biol, 2020). Downregulating or inhibiting Haspin could thus result in increased H3K4me3 levels and TFIID retention at the chromatin, at least at some genes (ext. fig. 5 and Discussion part P9, L40-42). Could the authors briefly discuss this point and the contribution of mitotic chromosome condensation itself in preventing readers to access histone marks?

Please also extend a bit the discussion on the direct phosphorylation of readers during M since it seems to be the main alternative mechanism to the methyl-phospho switch.

The first sentence of the introduction (P 3; L 2-4) is a statement that gene expression is regulated by 3D-genome architecture, which has been demonstrated in specific cases. I would tone this sentence down because, chromatin loops apart, a general link between transcription and higher order 3D-genome has not yet been established yet (Rowley et al, Nat. Genet., 2018 or Misteli, Cell, 2021).

Reviewer #1 (Remarks to the Author):

Summary of the main findings of the study.

The study provides compelling evidence that the methyl-phos switch mechanism does not function in vivo and is unlikely to be involved in widespread transcriptional repression in cells during mitosis. Therefore, this work indicates that H3T3ph is implausible to be responsible for transcriptional downregulation during cell division. For this reason, this work presents significant information for the field of mitosis and by extension for general cell biology research shedding new light on the mechanisms that regulate gene expression in human cells.

Strengths and limitations of this work.

The conclusions that this work presents are relevant for enhancing our knowledge about gene expression regulation. The main strength of this work lies in the comprehensive use of multiple in vivo and in vitro methodologies such as live imaging and IF, ELISAs on chemically synthesized peptides, Chip-seq, CIDOPseq, Kinase assays, and genome-wide distribution analysis of H3T3ph and H3K4me3. The work is furthermore complemented by different approaches to study H3T3ph and H3K4me3 related proteins: Haspin kinase (responsible for H3T3 phosphorylation) and several H3K3-reading proteins (TAF3, TAF5, ING2, Dido3 and ING2).

The conceptualization of this work is impressively thorough, incorporating a wide range of techniques and approaches to address the research question at hand. The results are neatly presented and properly discussed. Although no clear limitations were identified, some suggestions for improvement are provided below.

Comments and suggestions:

Introduction:

Please indicate which histone marks are critical for the sentence mentioned in lines 39-41 page 3 in relation to BRD4 bookmarking (references 32, 41, 42).

The binding patterns of BRD4 to acetylated H3 and H4 histones are complicated and hard to summarise concisely. Further details can be found in the cited references. As this is not central to the paper, we would rather not expand on this here, if the reviewer agrees.

Suggestion: In page 4 lines 6-7 authors might wish to cite some of the recent reports about Haspin activity during meiosis, as its role in H3T3 phosphorylation is conserved in both types of mammalian cell division (mitosis and meiosis). Examples of Haspin implications in meiosis and their relationship with histone modifications during meiosis are DOI: [10.1242/jcs.259546](https://doi.org/10.1242/jcs.259546), DOI: [10.1091/mbc.E16-12-0850](https://doi.org/10.1091/mbc.E16-12-0850), DOI: [10.1242/jcs.158840](https://doi.org/10.1242/jcs.158840) or DOI: [10.1242/jcs.189340](https://doi.org/10.1242/jcs.189340).

We agree that it is reasonable to mention the role of Haspin in meiosis, so we have included the earliest relevant reference (DOI: [10.1242/jcs.158840](https://doi.org/10.1242/jcs.158840)). (We already have a large number of references so we hesitate to include more).

Results:

This reviewer has some specific major and minor suggestions, which are explained below in the same order of appearance of the results section:

- Page 5 Lines 18-21

In Figure 1 panel B, why T3ph is not included? (major)

Equally to panel A, where anti-H3T3ph antibody B8634 is used against peptides carrying H3K4 methylation, panel B should indicate if anti-H3K4me3 antibody C42D8 detects peptides carrying H3T3ph alone. The graph includes H3K4me1, K4me2 or K4me3 in addition to H3T3ph, but it does not include H3T3ph alone.

The binding of the C42D8 anti-H3K4me2/3 antibody to H3T3ph peptides was already included in Figure 1D.

- Page 5 line 45 (minor)

For Extended Data/ED-Figure 2C, the text reads that “we also confirmed that the level of H3T3ph was largely preserved in preparations of sonicated chromatin from mitotic cells compared to whole HeLa cell lysates”, however all the figures and the material and methods section refers to this as

“sheared chromatin”. This reviewer is aware that the material methods sections explains that shearing protocol is done by sonication (lines 32-33 page 16) but please unify and mention the same terminology in lines 45-46 of pag 5.

We have corrected this to unify the terminology.

- Page 6 line 28 (minor)

For Figure 3, the text reads “the detected H3T3 kinase activity was strongly inhibited by a Haspin inhibitor”. As this is the first time that the inhibitor is mentioned in the text, please include here the specific information: indicate that inhibition is done with 5-iodotubercidin, in which H3T3ph is undetectable (cite references 57,65,66). This information is included for the first time in further pages of the original manuscript (in page 10). Please also indicate here in page 6 the concentration of use, and specify if it is the same one used for previous works of the group (reference 57).

We have specified the inhibitor on page 6 and added the relevant references. We have also added the concentration used to the legend of Figure 3.

Regarding the concentration of 5-Iodo inhibitor, the information provided in the manuscript is not clear. In page 16 line 7, it says that “cells were treated with 1 μ M 5-iodotubercidin for 2 h”. In contrast, page 17 line 20 says that “Where indicated, 5-iodotubercidin was included at 10 μ M.”. Please clarify the concentration used for all the experiments in this work: 1 μ M or 10 μ M.

The information on page 16 is in the section “Cell culture” and so refers to the concentration of iodotubercidin used during cell culture (1 μ M). The information on page 17 is in the section “In vitro kinase assays” and so refers to the concentration of iodotubercidin used during in vitro kinase assays (10 μ M). For the avoidance of doubt, we have added the concentrations used to the relevant legends of Figure 3 and Supplementary Figure 6.

- Page 6 line 37 (recommendation)

This reviewer believes that the schematic figure cited in line 28/page 6 (red box of ED-Figure 4) is useful and helps the understanding the relevance of the study. As a recommendation, a similar image, or a more detailed version summarizing the main conclusions of the work, could be included as Figure 8 within the discussion section. Incorporating this figure will surely bolster the overall message conveyed by this publication.

As recommended, we have moved this schematic to new Figure 8.

- Page 7, lines 8-10 (major)

Why does the Figure 5 not include quantification of endogenous TAF3 in the presence (Wild Type) or absence (Haspin KO) of H3T3ph in HeLa cells?

Live imaging results obtained for GFP-TAF5 in Hela and USO2SO are very robust. However, although the IF images of the endogenous protein are very representative, this reviewer believes that Figure 5 will benefit from a quantification analysis. This could be done by quantitative analysis of the relative signal intensities making sure that the acquisition time was fixed for all acquired images, and the quantification was only performed using the original unmodified images. Among other approaches, measuring the integrated fluorescence density in individual nuclei using ImageJ by creating a binary mask with the DAPI staining in a minimum of 10 cells per stage, could be an efficient approach, but author can choose another if they prefer.

As requested, we have now quantified the loss of endogenous TAF3 from chromosomes in mitotic cells from these IF experiments, and we do not see any significant difference between wild type and Haspin knockout cells (new Figure 5B). We used the approach outlined by the reviewer (which is essentially the method we used for live imaging), for 10 mitotic and 20 interphase cells per condition.

Secondly, as far as this reviewer understands, all the experiments of this work are done in HeLa cells, including GFP-TAG5 life images for WT and Haspin KO Hela cells. However, to study TAF5 under H3T3ph absence, Haspin inhibition with 5-Iodo has been done both in Hela and also in U2SO cells. Please, explain why TAF3 is the only protein addressed in both cell lines, in case there is a particular reason for it. In this regard, I would recommend a change in order of the figures for a better organization of the results: images of HeLa cells of ED-Figure 5 should be moved to ED-Figure 4 panel

C complementing the other results for HeLa Cells in panels A and B, and consequently, the only experiment done with U2Os cells could be separated and moved to ED-Figure 5.

We focus on TFIID in this work because the most extensive previous work on the H3T3ph-K4me switch used this example (Varier et al. 2010; ref 49). Varier et al. carried out GFP-TAF5 imaging in U2OS cells, so we wanted to check that the different results we obtained in HeLa cells were not due to cell line differences. Therefore, we analysed the effect of Haspin inhibition on the chromosomal localization of GFP-TAF5 in fixed U2OS cells (Supplementary Figure 6A, previously ED-Figure 4C) and, in response to reviewer 2, we have now also directly examined GFP-TAF5 behavior in live and fixed U2OS cells following Haspin RNAi (Supplementary Figure 5B, C). We confirm that we obtain the same results in both cell lines (that Haspin does not influence bulk displacement of TFIID from mitotic chromosomes).

We considered the reviewer's suggestion for rearranging figures, also in the light of the new data added. The logic here is that we discuss the effect of Haspin KO, Haspin RNAi and Haspin inhibition on TAF5 first, then the effect of Haspin loss on TAF3, rather than talking about one cell line then the other. We hope the changes to the order of the figures (which matches the order they are mentioned in the text) make sense. Supplementary Figure 5 is now devoted to GFP-TAF5 (part A in HeLa cells, and parts B and C in U2OS cells), while Supplementary Figure 6 is devoted to Haspin inhibitor results for both GFP-TAF5 and then GFP-TAF3 in U2OS and HeLa cells.

- Page 8 lines 2-11 (major)

Please indicate in the text that Figure 7 and ED-Figure 8 are both presented because they show IF of the same proteins but with different fixations: methanol for Figure 7 and formaldehyde for ED-Figure 8. Why is the reason behind this dual approach? Please explain this in the results section.

The reason for the dual approach is mentioned on page 7, lines 4-5: "to ensure they [the results] were not influenced by formaldehyde fixation artefacts". We have now added to the main text description of Figure 7 and ED-Figure 8 (now Supplementary Figure 9) on page 8 to make the use of different fixation conditions clearer here too. If this is insufficient, we are happy to expand on this further. In short, formaldehyde may effectively "pull" proteins that bind with weak affinity off chromatin in mitosis, perhaps by cross-linking them to cytosolic components as the chemical diffuses into the cell from the outside, as described in the cited reference 7. This is unlikely to happen with methanol fixation, so we show results with both methods, as well as with fixation-free live imaging.

Regarding this matter, the outcomes obtained for ING2 and LSD1 vary noticeably depending on whether methanol or formaldehyde was used for fixation. On the one hand, ING2 shows different patterns in WT cells if IF is done after methanol (Figure 7) or formaldehyde (ED-Figure 8) fixation. When fixed with the latter, ING2 shows a clear perichromatin localization in WT cells at telophase, whereas in Haspin KO it remains diffuse at the cytoplasm at that stage (ED-Figure 8). Concerning this issue, it is worth noting that ING proteins have been reported to interact with Lamin A in the nuclear envelope, authors might take this into consideration (as an example Han et al 2008 <https://www.nature.com/articles/ncb1792>). On the other hand, Figure 7 (methanol fixation) shows equal signals for LSD1 in WT and Haspin KO cells. In contrast, ED-Figure 8 (formaldehyde fixation) shows that LSD1 is reincorporated to the perichromatin at telophase in WT, but not in Haspin KO. This might be related to the appointed role for LSD1 in the histone demethylation as a regulatory mechanism linking the chromatin state with the nuclear envelope reorganization at the end of mitosis (Schooley et al 2015m DOI: [10.1242/jcs.173013](https://doi.org/10.1242/jcs.173013)). Authors are encouraged to clarify the reasons behind the divergent results obtained through different fixation techniques in HeLa cells at telophase, and if desired, discuss and explore potential reasons for the different outcomes observed for ING2 and LSD1 in WT versus Haspin KO (ED-Figure 8).

These potential differences in ING2 and LSD1 behaviour are interesting, but we do not believe that there is a consistent difference in the localization of these proteins between wild type and Haspin KO cells in either fixation condition. The wild type anaphase/telophase cells shown in Supplementary Figure 9 (previous ED-Figure 8) are more rounded up than the corresponding KO cells, which makes the cytosolic ING2 look more intense and closer to the chromosomes than in the more flattened KO cells. To illustrate this point, the Reviewer Figure below shows an alternative pair of images from an independent formaldehyde-fixation experiment where both WT and KO cells show very similar localisation patterns. A similar argument applies to LSD1. Therefore, we do not think there is sufficient evidence to pursue the line of investigation suggested by the reviewer (who of course had to rely on single images in the submitted manuscript).

Reviewer Figure. Immunofluorescence microscopy (with formaldehyde fixation) for DNA (blue), ING2 (green), and CENP-C (centromeres, red) in wild type and Haspin knockout HeLa cells.

Discussion: The discussion encompasses all pertinent previous studies on the topic and provides a robust rationale for the findings. That being said, I would like to offer a couple of suggestions:

- This reviewer is aware of space and reference number limitation, but some additional discussion about the transcription at centromeric regions will benefit results obtained from Figure 6B. This could be included in relation to lines 10-16 in page 10. As an example, reviews from Hong Liu (2016) doi: [10.1080/21541264.2015.1127315](https://doi.org/10.1080/21541264.2015.1127315) , or Chan and Wong (2012) DOI: [10.1093/nar/gks921](https://doi.org/10.1093/nar/gks921) are interesting to revise for original references. More specifically, previous works suggesting that the centromeric transcription helps maintain centromeric proteins at centromeres and about the essential role of RNA Polymerase II at the centromeres during mitosis are relevant (Chan et al 2012, DOI: [10.1073/pnas.1108705109](https://doi.org/10.1073/pnas.1108705109), Hall et al 2012 DOI: [10.1007/s10577-012-9297-9](https://doi.org/10.1007/s10577-012-9297-9)).

We note that the results in Figure 6B (and elsewhere) show conventional annotated TSSs of well-defined genes in non-repetitive, largely euchromatic, regions excluding centromeres (ie GENCODE annotated protein and non-coding RNA genes). Our work does not include analysis of the putative and still ill-defined TSSs in the satellite DNA of centromeres. Furthermore, the highly repetitive nature of these regions means that the location of H3K4me and H3T3ph within centromeres is difficult to determine. For this reason, while very interesting, we think it is best to avoid discussion of centromeric transcription in mitosis here as we do not want to imply that we make any conclusions about these regions.

- As previously commented for Figure 7 and ED-Figure 8, discussion about the role of LSD1 in nuclear reassembly could also be interesting. In this regard, as mentioned above, previous results indicated that demethylation of histone H3 by LSD1 is involved in the regulation of nuclear reassembly at telophase (Schooley et al 2015, DOI: <https://doi.org/10.1242/jcs.173013>).

While interesting, we believe these points are beyond the scope of this work and, for the reasons outlined in the relevant point above, we think it is premature to discuss them at this stage.

Reviewer #2 (Remarks to the Author):

The manuscript from Harris et al investigates the interplay of histone H3T3ph and histone H3K4me3 modifications during mitosis. The work offers a reconsideration of previously established findings, where H3T3ph was found to block the binding of reader proteins to H3K4me3 in vitro and the H3T3ph kinase Haspin was shown to promote eviction of H3K4me3-dependent transcription factors from chromatin in cells. In the current study, the authors build on prior results using histone peptides and demonstrate that H3K4me3 can decrease H3T3 phosphorylation in the context of nucleosomes. Importantly, they used ChIP-seq to map H3K4me2/3 and H3T3ph in mitotic cells and found that H3T3ph is not enriched at H3K4me2/3 peaks at gene promoters. Using CIDOP-seq, they further show that the H4K3me2/3 binding PHD domain of TAF3 can still bind to H3K4me3 in mitotic chromatin, suggesting that other signals evict the TFIID complex from chromosomes in mitosis. In line with this, but in opposition to prior results (Varier et al, EMBO, 2010), they

find that Haspin depletion or inhibition does not affect the eviction of TAF3 and other H3K4 readers from mitotic chromosomes. The authors conclude that H3T3ph is not a “switch” in vivo.

The main advance of this paper is the demonstration that H3K4me3 and H3T3ph are not overlapping modifications on mitotic chromatin. Thus, although previous work from different labs have shown that H3T3ph is antagonistic towards H3K4me2/3 readers in vitro, this work seeks to make the point that this is not physiologically relevant to the eviction of these readers during mitosis in vivo. These findings are important and will be relevant to researchers in both the transcription and cell division fields. However, much of the work here is either confirmatory or in direct conflict with previous results and does not provide enough evidence to rule out a role for H3T3ph in the eviction of H3K4 readers, in particular TAF3 of the TFIID complex. To be considered for publication in Nature Communications, my opinion is that further evidence is required to support the conclusion that Haspin and H3T3ph do not play a role in the eviction of H3K4 readers.

1) I recommend that Figures 1 and 3 be moved to the supplementary materials section due to the fact that they are mainly supportive findings or have been previously published in some form. I also recommend that Figures 4 and 5 be merged and some aspects be moved to the supplement because they are essentially covering the same point.

We understand the reviewer’s points here, but we think in general it is best to reduce the number of times a reader has to refer to supplemental material. Although antibody characterization may frequently be relegated to the supplement, we think, in the case of histone modifications, it is hard to overstate the importance of proper antibody testing (see refs 61, 62). Indeed, the reviewer later makes the point that the fact that the H3K4me antibody recognises both H3K4me2 and me3 is important for interpretation of the work. Of course, if necessary, we will move Figure 1 to the supplement, but we deliberately chose to include these experiments in the main figures to emphasize their importance.

We would also prefer to leave Figure 3 in the main paper. The results here go further than previously published work, and form an important part of the argument made in the paper. We now refer to the lengthened dose-response shown in Figure 3A in both the Results and Discussion section, and it is probably better that a reader does not have to refer to the supplement twice to understand these points.

In response to reviewer 1, we have expanded Figure 5, and so we prefer to keep Figure 4 and 5 separate to avoid the figure becoming too large.

2) One of the main points of the paper is that H3T3ph signals are not enriched at H3K4me2/3 peaks in mitotic cells enriched for prometaphase to metaphase. However, since most of the H3T3ph signal is enriched at repetitive centromeres and the peri-centromeric chromatin due to removal of cohesin from the arms by the prophase pathway (the PDS5 subunit of cohesin helps to enrich Haspin), it is unclear how robust the H3T3ph signals are on bulk chromatin and whether the anti-correlation between H3T3ph and H3K4me2/3 is relevant (the amount of H3T3ph is quite low). According to their results, TAF3 is evicted early on during prophase, and so this ChIP-seq dataset may not reflect an accurate snapshot of the chromatin landscape during prophase where H3T3ph has been reported to be more enriched on chromosome arms. To get an idea of the robustness of the H3T3ph signal, I suggest that the authors perform ChIP-seq on Haspin KO cells. To get a better snapshot of cells in a prophase-like state, the authors can deplete WAPL, which forces cohesin and RNA Polymerase II onto the chromosome arms (Perea-Resa et al, Mol Cell, 2020), and would likely increase Haspin-dependent H3T3ph in these regions. With more robust H3T3ph signals in the context of increased RNA Pol II, I feel that the authors can make a stronger case for their claim that “H3K4me3 hinders adjacent H3T3ph deposition in cells”.

The reviewer makes a valid point here. Indeed, in common with essentially all similar work in the field, we are analysing cells in a drug-induced mitotic state approximating late prometaphase, when H3T3ph on chromosome arms is low relative to centromeres, and it would be interesting to know the distribution of H3T3ph in prophase when it may be more biased towards the chromosome arms.

We agree that the extent of H3T3ph signal on chromosome arms in our studies is difficult to ascertain. Because interpretation of ChIP-seq depends on *enrichment* at specific chromosomal loci, this is a problem inherent in the approach when assessing an epitope that may be fairly evenly distributed across large chromosome regions. H3T3ph ChIP from asynchronous cells yielded too little DNA to sequence by our method (as stated in the text), and the same will almost certainly be true for mitotic Haspin KO cells. Nevertheless, the fact that a dip in H3T3ph signal can be seen at H3K4me3 peaks genome-wide (Figure 2D; new Supplementary

Figure 4B) is indicative that there is H3T3ph detectable on arms, particularly because well-known ChIP-seq artifacts tend to *increase* signals at promoter regions. However, precisely because of the lower signal on chromosome arms, we focus our analysis on peri-centromeric regions containing well-annotated genes where H3T3ph is clearly strong. To reinforce this, we have now included plots of H3T3ph enrichment across H3K4me3 peaks in these centromere-proximal regions in Supplementary Figure 4A (as we already had done for TAF3 CIDOP-seq in Figure 6).

The idea to increase Haspin-dependent H3T3ph on chromosome arms by WAPL depletion is an interesting one. However, there was not an obvious increase in H3T3ph on chromosome arms in the Perea-Resa et al. study (see their Figure 5D). We note also that WAPL depletion will alter progression through mitosis, and that we would still not know if such artificial Haspin recruitment reflected the normal prophase situation. It is true that we extrapolate the results from centromere-proximal regions in nocodazole-arrested mitotic cells to the potential effects on chromosome arms (which may be more relevant in prophase), but this view is supported by additional data, including the fact (i) that H3T3ph reaches its highest levels at prometaphase centromeres, and this is much higher than the low levels seen on chromosome arms in prophase (at least by IF), and so peri-centromeric regions represent a stern test of the ability of H3K4me3 to antagonise H3T3ph, (ii) that H3K4me3 directly hinders Haspin activity towards H3T3 *in vitro*, (iii) that bulk displacement of a number of H3K4me3-readers from all regions of mitotic chromosomes is unaffected by loss of Haspin activity in immunofluorescence and live cell imaging experiments. We have also altered the text to make the description of results in centromere-proximal versus arm regions clearer.

3) It was previously shown that Haspin overexpression in interphase leads to TAF3 and TFIID eviction and abrogation of TFIID transcription in U2OS cells (Varier et al). These results provided strong evidence that outside of mitosis, H3T3ph is sufficient for eviction. In the same paper, it was also shown that knockdown of Haspin by RNAi hampered eviction of the TFIID subunit TAF5 in mitosis. One potential view of these results in light of the current manuscript is that H3T3ph can aid in TFIID eviction, but it is not the primary signal. The role of H3T3ph in this regard may be cell-type or locus specific, as the prior work was performed in U2OS cells and the current work utilized HeLa cells. To investigate this further, the authors should perform live imaging with GFP-TAF3 in U2OS cells depleted of Haspin. In addition, the authors could utilize the LacO-LacI system to target H3T3ph and H3K4me3 to specific LacO repeats in mitotic U2OS and HeLa cells to determine the interplay between these marks and TAF3 recruitment at a defined locus away from centromeres in different cell types.

The previous version of the paper already included the results of treating GFP-TAF5-expressing U2OS cells with the Haspin inhibitor 5-iodotubercidin. In response to this comment, we have now also included the results of Haspin RNAi in this cell line, which also showed no change in dissociation from chromosomes in mitosis. These experiments were performed both with living cells (Supplementary Figure 5B), to eliminate the potential for fixation artefacts, and in fixed cells (Supplementary Figure 5C), to demonstrate that H3T3ph was efficiently prevented in the majority of cells by Haspin RNAi.

The idea behind using the LacO-LacI system to target histone modifications to a specific locus that can then be engineered in controllable ways is a good one. However, this system would be complicated to set up, as it might need fine control of the sequence of H3T3ph and H3K4me3 deposition, as well as ChIP analysis across the locus to confirm the exact relative locations of the two marks. Furthermore, H3K4me3 is only one contributor to the chromosomal activity of TFIID, which recognises a number of features at promoters, and seems unlikely that H3K4me3 alone would be sufficient to recruit TFIID to the artificial locus in cells. Displacement mechanisms such as TFIID phosphorylation by Cdk1 would also be at play in the cellular context, and it would require extensive analysis to define the nature of these additional regulatory pathways on TFIID and its subcomponents (like TAF3) to generate suitable separation-of-function mutants.

4) Please describe how the curves were fit in Figure 3. Also, the curve in Figure 3A does not go to completion for the H3K4me3 nucleosomes. Why is 1 nM haspin chosen as 100%? This is an important point since Haspin can be enriched on chromatin by cohesin, and therefore could exist in locally high concentrations. These local concentrations could overcome the inhibitory effects of H3K4me3, and it would be important to understand the relationship between Haspin concentration and phosphorylation in the presence of H3K4me3. This is relevant to the discussion on page 9, lines 34-42, where the authors cast doubt on prior results that utilized Haspin overexpression in interphase cells.

This is a very good point. In fact, the original assay shown in Figure 3A included a 10 nM Haspin condition. Early drafts of the manuscript included these data because they support the idea, mentioned in the Discussion

and by the reviewer, that high levels of Haspin can overcome the inhibitory effect of H3K4me3. However, the results were not included in the final figure because 10 nM was a significantly higher concentration than the others used in the experiment (all ≤ 1 nM), which made visualisation of the data at lower concentrations harder. However, we have now reinstated them. Curve fitting is done using hyperbolic dose-responses in Prism, but we do not draw strong conclusions from the exact nature of the curves. Theoretically, the dose-response to increasing Haspin concentration would likely be linear at low concentrations, but not at higher concentrations as substrate becomes limiting.

5) It is important to point out in the manuscript that prior work has shown that H3K4me2 is not very inhibitory towards Haspin-dependent H3T3 phosphorylation. This is relevant because, as the authors nicely show in Figure 1, their antibody recognizes both H3K4me2 and H3K4me3 equally. This is particularly relevant for the sentence in line 38, page 6.

This is a reasonable point, and we have rephrased the relevant line on page 6 (now line 45) to accommodate this fact, and we have also explicitly described the extent of inhibition on lines 27-29.

Reviewer #3 (Remarks to the Author):

This study by Harris and colleagues challenges the model by which a methyl-phos switch is at least partially responsible for the release of transcriptional activators from mitotic chromatin, which contributes to transcriptional attenuation at mitosis. The focus of the study is on the methyl-phos switch H3K4me2/3 – T3Ph. Phosphorylation of H3T3 by Haspin prevents the recognition of H3K4me3 by the reader TFIID/TAF3, which in turns reduces transcription.

This model is currently based on *in vitro* data and the authors investigated it further using a combination of elegant *in vitro* and *in-cell* approaches. Interestingly, ChIPseq showed that H3T3ph is not present at promoters featuring H3K4me3 during M phase. At the same time, as expected, TFIID/TAF3 occurrence at these promoters decreased compared to interphase. These data thus exclude the possibility of a methyl-phos switch mechanism to exclude TFIID from mitotic chromatin. Although the authors do not have data pointing at an alternative mechanism, this study is very well conducted, very well balanced between *in vitro* and *in-cell* experiments and very well presented and written. The results are clearly displayed and relevant controls included. This study will be of interest to a broad readership. Nevertheless, I listed some points below that should be addressed before publication. I hope that they will be helpful.

Major

1) The decreased presence of TFIID/TAF3 at mitotic chromatin vs interphase chromatin is not obvious in Figure 6 and Ext. Figure 7. Especially when looking at heat maps and genome browser snapshots. The authors should adjust the scales of the density plots so that we could clearly see that TFIID presence at mitotic chromatin is decreased compared to interphase (same scale adjustments apply for H3K4me3 metagenes in Figure 2B). Is the amplitude of this decrease uniform genome-wide or stronger at specific subsets of genes. Does it correlate with transcriptional attenuation? (see point 2-4)

Please note that CIDOP-seq is a technique to analyse the *in vitro* binding of a factor to chromatin isolated from cells. TAF3-PHD binding in this assay is non-physiological, and does not reflect the location of TAF3 in mitotic cells (we've introduced some changes in the paper to make this clearer).

It is also important to note that our ChIP-seq and CIDOP-seq data are not quantitative in a way that allows comparisons of peak heights between interphase and mitosis. We considered the inclusion of spike-in controls, but this would not fully address this problem either, because the condensation state of chromatin is so different in interphase and mitosis that we could not confidently interpret the significance of any changes in peak heights (also note that different sonication protocols were required for interphase vs mitotic chromatin). Because of this, we quite deliberately chose to show the graphs in Figure 2, 6, and Supplementary Figure 8 (previous ED-Figure 7) in a way that does not visually imply a difference in peak heights. We worry that adjusting the scales in the way the reviewer mentions would actually have the potential to be more misleading. For these reasons, throughout the manuscript, we are careful to draw our conclusions only from the *intra*-sample differences in H3T3ph, H3K4me, and TAF3-PHD locations along chromosomes within identical preparations of chromatin.

We would also point out that the key result here is that, by CIDOP-seq, we *can* see TAF3 binding to mitotic chromatin near centromeres *in vitro*, even when this wouldn't happen *in vivo*, and even though we find H3T3ph is present at high levels (surrounding, but not coincident with, H3K4me2/3) in exactly the same chromatin preparation using ChIP-seq. The important point from this is that the presence of H3T3ph cannot explain loss of TFIID from mitotic chromosomes.

We find no evidence for selective differences in the binding of TAF3-PHD to different H3K4me2/3 peaks in interphase versus mitotic chromatin preparations: (i) the binding of TAF3-PHD in H3T3ph-containing pericentromeric regions is similar to that on chromosome arms with low H3T3ph (Figure 6; Supplementary Figure 8D); and (ii) the correlation between peak enrichment of TAF3-PHD at interphase and mitotic sites by CIDOP-seq is similar or greater than that between interphase and mitotic peaks of H3K4me2/3 by ChIP-seq (Supplementary Figure 8A), even though the CIDOP-seq data is noisier (Supplementary Figure 8B, presumably because TAF3-PHD binding to chromatin is weaker than that of H3K4me2/3 antibodies), and the retention of H3K4me2/3 in mitosis relative to interphase is well established (Javasky et al. 2018; ref 30). This provides strong evidence that the pattern of TAF3-PHD binding to H3K4me2/3 on mitotic chromatin *in vitro* is as similar to interphase chromatin as is observed for H3K4me2/3 itself. Again, the mitotic binding of TAF3-PHD observed by CIDOP-seq is non-physiological; we deliberately use it as an experimental tool to explore the influence of H3T3ph in a simplified system.

2) Genome-wide H3K4me3 ChIP intensity doubles between interphase and M (Figure 2). As for TFIID above, is it a homogeneous increase and does it correlate with gene expression variations?

Although H3K4me3 may increase during M at bivalent genes, I would expect this mark to remain relatively stable from interphase to M in HeLa cells. The increased mitotic presence of H3K4me3 observed by the authors could actually favor retention of TFIID and sustain transcription during M, at least at some genes like housekeeping genes. Or maybe it is only bookmarking and H3K4me3 is not accessible to TFIID due to chromatin compaction. The authors should comment on this and perform transcriptomics (see below).

As described above, we cannot make such quantitative arguments about changes in enrichment levels between different chromatin preparations, especially between interphase and mitosis. Indeed, as the reviewer surmises, the lack of change in H3K4 methylation between interphase and mitosis has been confirmed by other work, including in HeLa cells by Liang et al. (2015; ref 15), and in another careful study in HeLa cells using a combination of ChIP-seq and quantitative mass spectrometry (Javasky et al. 2018; ref 30). The correlation between the pattern of mitotic and interphase peak enrichment in our study is also high, showing that there is no major change in H3K4me2/3 distribution in mitosis, as expected (Supplemental Figure 8A). Of course, we cannot rule out a low number of gene-specific changes in H3K4me3 levels, but we consider this beyond the scope of our study which focuses on the role of H3T3ph.

As the reviewer points out, because H3K4me3 remains present during mitosis, and H3T3ph is generally not co-located with it, H3K4 methylation could certainly still retain some TFIID at some promoters. Indeed, this is one implication of our work that we cover in the Discussion.

3) Indeed, gene expression data would benefit this study. Gene expression in interphase and mitosis should be assessed (or collected from a previous study if possible) and integrated with H3K4me3/T3ph ChIPseq and TFIID CIDOP-seq data. These data will bring more depth to current conclusions, which right now seem to draw a uniform model that would apply to most genes and promoters in M phase while previous studies indicated differential regulation of subsets of genes during mitosis (e.g. Palozola et al, Science, 2017).

To illustrate my point, the authors mention that T3ph is “rarely” found at promoter were K4me3 is found. What is the actual proportion of promoters having both T3ph and K4me3? Then what is the status of TFIID at such promoters and what are the expression levels of these genes compared to interphase? Maybe their mitotic downregulation is stronger than at genes were T3ph is not found at the promoter with K4me3?

Because of the nature of our ChIP-seq data, we are not convinced that we can make definitive calls of individual H3K4me3 peaks that have H3T3ph. Nevertheless, to address the question about the nature of any promoters with H3K4me3 that also have H3T3ph, we collected from our data all genomic bins in which both H3T3ph and H3K4me2/3 were called as positive. We then generated a list of all genes where “double positive” H3T3phK4me2/3 bins were found within a 2 kb window surrounding TSSs, which amounted to 0.1% of annotated genes, or 0.3% of genes with an H3K4me2/3-containing promoter. As expected from the

distribution of H3T3ph across chromosomes, the majority of these genes were in pericentromeric regions. We did not, however, detect any enrichment in particular gene ontologies or KEGG pathways. Furthermore, to more directly address the reviewer's question, we made use of nascent RNA-seq data from mitotic and asynchronous HeLa cells obtained using synchronisation protocols very similar to our own (Liang et al, 2015; ref 15). We did not detect any statistically significant enrichment of genes that are relatively high- or low-expressed in mitotic versus asynchronous cells within our list of genes that have both H3T3ph and K3K4me2/3 at promoters. Therefore, we could not find evidence of a correlation of H3T3ph at specific H3K4me3 promoters with expression pattern during the cell cycle, nor with a particular functional category of genes. We did not intend to "draw a uniform model". Our main argument is that H3T3ph is not responsible for the bulk of TFIID displacement in mitosis. In fact, this opens up the possibility of non-uniform models, in part because there is no global displacement of TFIID by H3T3ph, and the possibility that H3K4me readers are retained at a subset of promoters is mentioned in the Discussion. However, we acknowledge that perhaps this was not sufficiently clear in the manuscript, and we have now emphasised the lack of a role of H3T3ph in overall/bulk displacement of H3K4me readers, and point out that differential regulation of H3K4 reader phosphorylation, for example, could still provide locus-specific control.

Together, the points above could strengthen the conclusion that TFIID release from mitotic chromatin and decreased genes expression are not linked to T3ph. They could also reveal subsets of genes at which transcription, histone marks and TFIID recruitment do not correlate fully with the model proposed by the authors, which is likely since ext. fig. 5 indicates that TAF3 substantially remains on mitotic chromatin after Haspin inhibition.

We think the anti-correlation of DNA and GFP-TAF3 by immunofluorescence in ED-Figure 5 (now Supplementary Figure 6B) is striking (and this is even more obvious when looking at individual confocal slices). While of course this method cannot rule out retention at select foci, we do not think this experiment provides evidence for TAF3 retention on mitotic chromatin, particularly as GFP-TAF3 is overexpressed in these conditions.

I am just wondering whether, although not predominant in cells, the methyl-phospho switch could still coexist with this new model, at specific subsets of genes, to exclude TFIID and decrease transcription. This is a reasonable idea, and in fact our original hypothesis going into this study was that selective deposition of H3T3ph at different loci in mitosis would cause differential displacement of H3K4 readers by the methyl-phospho switch. However, we couldn't find any evidence for switch function in cells and we find very few, if any, genes where there is evidence of H3T3ph coinciding with H3K4me3 at promoters (see point 3 above). Nonetheless, we have now made clearer in the manuscript that we cannot rule out a role for the switch at select loci.

Minor

ChIPseq in Fig. 2 and in vitro data in Fig. 3 are clear. But, is K4me3 preventing phosphorylation of T3 during mitosis? To establish a direct link, inhibition of H3K4 methylation could be done, maybe by downregulating the WDR5 protein like in Lauberth et al, Cell, 2013. One would presume that more T3ph will be detected at promoters upon downregulation of K4me3 in ChIPseq or ChIPqPCR experiments.

We did consider this experiment, but we failed to identify an efficient way to disrupt H3K4 methylation. Also, even if we found that H3T3ph did not move into promoters following loss of H3K4me, we reasoned that this wouldn't undermine the argument that H3T3ph is not responsible for displacement of most H3K4me readers from mitotic chromatin because we find that H3T3ph is depleted at H3K4me2/3 peaks.

TFIID binding is not influenced by the distance from the centromere, which suggests that high H3T3ph ChIP intensity around H3K4me3 peaks has no effect on gene expression. Still, I was wondering whether gene expression is differentially regulated between the pericentromeric H3T3ph broad peak, compared to chromosome arms during M phase. This would point at an influence of H3T3ph (or other marks like H3S10ph) on gene expression that is not related to a methyl-phospho switch or regulation of TFIID. The answers to this question could be gathered from results collected in point 3-4 above.

This is an interesting question, but it is different from the one we address in this study. Any such correlation would not distinguish which centromeric factors were responsible, and so would need extensive follow-up

studies with depletion of potential centromeric players such as Haspin, Aurora B, Bub1, heterochromatin proteins etc. In a preliminary analysis, we determined if genes in the cytogenetic bands immediately adjacent to all centromeres (which are high in H3T3ph – see Supplemental Figure 3A) are over-represented among the genes identified by Liang et al. (2015; ref 15) to be relatively high- or low-expressed in mitotic versus asynchronous cells. This did not reveal any such enrichment.

Preventing mitotic condensation of chromosomes in yeast triggers an increased occurrence of H3K4me3 and unscheduled gene expression (Ramos-Alonzo et al, PNAS, 2023). This is linked to Aurora B activity, which can be recruited by Haspin (Hadders et al, J Cell Biol, 2020). Downregulating or inhibiting Haspin could thus result in increased H3K4me3 levels and TFIID retention at the chromatin, at least at some genes (ext. fig. 5 and Discussion part P9, L40-42). Could the authors briefly discuss this point and the contribution of mitotic chromosome condensation itself in preventing readers to access histone marks?

We note that there is little evidence that Aurora (Ipl1) recruitment to centromeres is dependent on the Haspin homologues (Alk1 or Alk2) or H3T3ph in budding yeast. Also, in human cells, overall chromosome condensation in mitosis is not altered by Haspin or Aurora B loss (though of course we cannot rule out a subtler effect). So, we think detailed discussion of this particular point would be rather speculative. Nevertheless, it is certainly reasonable to mention the possible role of condensation in controlling access to mitotic chromatin, which we now do in the Discussion.

Please also extend a bit the discussion on the direct phosphorylation of readers during M since it seems to be the main alternative mechanism to the methyl-phospho switch.

We have added an additional sentence to the Discussion highlighting the likely role of Cdk1-Cyclin B, and how differential phosphorylation might give locus-specific control.

The first sentence of the introduction (P 3; L 2-4) is a statement that gene expression is regulated by 3D-genome architecture, which has been demonstrated in specific cases. I would tone this sentence down because, chromatin loops apart, a general link between transcription and higher order 3D-genome has not yet been established yet (Rowley et al, Nat. Genet., 2018 or Misteli, Cell, 2021).

A reasonable point, and we have reworded this as requested.

REVIEWERS' COMMENTS

Reviewer #1 (Remarks to the Author):

Authors have improved the manuscript answering to most my concerns and implementing most of the suggestions. Specifically, they have made the following changes:

- They have introduced some information about the role of Haspin during meiosis, complementing the already introduced references about mitosis.
- They have unified the terminology when referring to sonicated chromatin and sheared chromatin.
- They have specified references and details for the inhibitors used.
- They have specified the adequate concentrations of 5-iodotubercidin when used for in vitro kinase assays (10 μ M) or for cell culture (1 μ M).
- They have moved red box of ED-Figure 4 to Figure 8 as suggested.
- They have quantified the loss of endogenous TAF3 from chromosomes in mitotic cells from these IF experiments as suggested. This data reinforces the previous results.
- They have now also directly examined GFP-TAF5 behaviour in live and fixed U2OS cells following Haspin RNAi, as suggested by reviewer 1 and 2.
- They have clarified why Figure 7 and ED-Figure 8 are both presented in this work, because they show IF of the same proteins but with different fixations.

Pending suggestions: this reviewer still has some answers, but the response given by the authors are enough to move forward and accept the manuscript, as most of the important concerns are already answered.

-I don't see the in Figure 1 panel B if anti-H3K4me3 antibody C42D8 detects peptides carrying H3T3ph (bar colored in blue and labeled T3ph in the rest of the figure). The graph includes H3K4me1, K4me2 or K4me3, but it does not include H3T3ph alone. It does include H3T3ph (bar colored in blue in the rest of the figure) it in figure 1 panel D, binding to peptide and nucleosomal substrates, but I don't understand why it can't be included in panel B.

-This reviewer still has remaining doubts about the potential importance of the reincorporation of the LSD1 signal to the perichromatin at telophase in WT, whereas in Haspin KO it remains diffuse at the cytoplasm at that stage. I understand that my concerns might be based on the single images submitted in the original manuscript. However, I understand that it might exceed the scope of this work. Nevertheless, I encourage the authors to keep these questions open to be addressed in future works.

In summary, I would like to congratulate the authors for this excellent work and thank them for considering my suggestions.

Reviewer #2 (Remarks to the Author):

The authors have addressed the majority of my concerns, and I support publication in Nature Communications. Below are a couple of points that should be addressed.

In the abstract, the authors mention only H3K4me3 and not H3K4me2. As their antibody recognizes both me2 and me3 marks equally, it seems best that they use "H3K4me2/3" in their abstract where appropriate. Since H3K4me2 doesn't inhibit T3ph nearly as much as H3K4me3, I would suggest they modify the word "hinders" on line 8 to reflect this (e.g. anti-correlated?).

If space allows, it might be worth referencing a paper that demonstrated a role for the pre-initiation

complex in condensin loading during mitosis (PMID: 35293859) in the Discussion section, as it supports the notion that there is a function for residual GTFs in mitosis and that Haspin doesn't interfere with this.

Reviewer #3 (Remarks to the Author):

In this revised version of their study Harris and colleagues have addressed all of my questions and concerns satisfactorily and significantly clarified some aspects of their experimental procedures and interpretations of the results. These data support a new model in which H3T3ph is not anymore primarily responsible for the eviction of H3K4me3 readers like TFIID from mitotic chromatin. I have no further comments.

Reviewer #1 (Remarks to the Author):

Authors have improved the manuscript answering to most my concerns and implementing most of the suggestions. Specifically, they have made the following changes:

- They have introduced some information about the role of Haspin during meiosis, complementing the already introduced references about mitosis.
- They have unified the terminology when referring to sonicated chromatin and sheared chromatin.
- They have specified references and details for the inhibitors used.
- They have specified the adequate concentrations of 5-iodotubercidin when used for in vitro kinase assays (10 μ M) or for cell culture (1 μ M).
- They have moved red box of ED-Figure 4 to Figure 8 as suggested.
- They have quantified the loss of endogenous TAF3 from chromosomes in mitotic cells from these IF experiments as suggested. This data reinforces the previous results.
- They have now also directly examined GFP-TAF5 behaviour in live and fixed U2OS cells following Haspin RNAi, as suggested by reviewer 1 and 2.
- They have clarified why Figure 7 and ED-Figure 8 are both presented in this work, because they show IF of the same proteins but with different fixations.

Pending suggestions: this reviewer still has some answers, but the response given by the authors are enough to move forward and accept the manuscript, as most of the important concerns are already answered.

-I don't see the in Figure 1 panel B if anti-H3K4m3 antibody C42D8 detects peptides carrying H3T3ph (bar colored in blue and labeled T3ph in the rest of the figure). The graph includes H3K4me1, K4me2 or K4me3, but it does not include H3T3ph alone. It does include H3T3ph (bar colored in blue in the rest of the figure) it in figure 1 panel D, binding to peptide and nucleosomal substrates, but I don't understand why it can't be included in panel B.

The experiments shown in Figure 1B did not include H3T3ph peptides so, although we realise it might be simpler for the reader, we cannot include such data here. However, it is of course vital to know that the H3K4me2/3 antibody does not recognise H3T3ph, and we include these data in Figure 1D. We do not believe this has any substantive effect on the quality or interpretation of the data.

-This reviewer still has remaining doubts about the potential importance of the reincorporation of the LSD1 signal to the perichromatin at telophase in WT, whereas in Haspin KO it remains diffuse at the cytoplasm at that stage. I understand that my concerns might be based on the single images submitted in the original manuscript. However, I understand that it might exceed the scope of this work. Nevertheless, I encourage the authors to keep these questions open to be addressed in future works.

Of course we will bear this in mind, thank you.

In summary, I would like to congratulate the authors for this excellent work and thank them for considering my suggestions.

Reviewer #2 (Remarks to the Author):

The authors have addressed the majority of my concerns, and I support publication in Nature Communications. Below are a couple of points that should be addressed.

In the abstract, the authors mention only H3K4me3 and not H3K4me2. As their antibody recognizes both me2 and me3 marks equally, it seems best that they use "H3K4me2/3" in their abstract where appropriate. Since H3K4me2 doesn't inhibit T3ph nearly as much as H3K4me3, I would suggest they modify the word "hinders" on line 8 to reflect this (e.g. anti-correlated?).

These are reasonable points, and we have updated the abstract to state "H3K4me2/3" where appropriate, and have used "anti-correlates with" rather than "hinders".

If space allows, it might be worth referencing a paper that demonstrated a role for the pre-initiation complex in condensin loading during mitosis (PMID: 35293859) in the Discussion section, as it supports the notion that there is a function for residual GTFs in mitosis and that Haspin doesn't interfere with this.

We have added this reference to the Discussion.

Reviewer #3 (Remarks to the Author):

In this revised version of their study Harris and colleagues have addressed all of my questions and concerns satisfactorily and significantly clarified some aspects of their experimental procedures and interpretations of the results. These data support a new model in which H3T3ph is not anymore primarily responsible for the eviction of H3K4me3 readers like TFIID from mitotic chromatin.

I have no further comments.